# Computationally efficient mechanism discovery for cell invasion with uncertainty quantification

**Daniel J. VandenHeuvel** **, Christopher Drovandi** , **Matthew J. Simpson** *

School of Mathematical Sciences, Queensland University of Technology, Brisbane, Queensland, Australia

* matthew.simpson@qut.edu.au

## Abstract

Parameter estimation for mathematical models of biological processes is often difficult and depends significantly on the quality and quantity of available data. We introduce an efficient framework using Gaussian processes to discover mechanisms underlying delay, migration, and proliferation in a cell invasion experiment. Gaussian processes are leveraged with bootstrapping to provide uncertainty quantification for the mechanisms that drive the invasion process. Our framework is efficient, parallelisable, and can be applied to other biological problems. We illustrate our methods using a canonical scratch assay experiment, demonstrating how simply we can explore different functional forms and develop and test hypotheses about underlying mechanisms, such as whether delay is present. All code and data to reproduce this work are available at https://github.com/DanielVandH/EquationLearning.jl.

**Data Availability Statement:** All data and code available at https://github.com/DanielVandH/EquationLearning.jl.

**Funding:** CD recieved funding from the Australian Research Council (FT210100260) https://www.arc.

## Author summary

In this work we introduce uncertainty quantification into equation learning methods, such as physics-informed and biologically-informed neural networks. Our framework is computationally efficient and applicable to problems with unknown nonlinear mechanisms that we wish to learn from experiments where only sparse noisy data are available. We demonstrate our methods on a canonical scratch assay experiment from cell biology and show the underlying mechanisms can be learned, providing confidence intervals for functional forms and for solutions to partial differential equation models believed to describe the experiment.

This is a *PLOS Computational Biology* Methods paper.

## 1 Introduction

Estimating parameters from biological experiments using experimental data is a difficult problem. Biological data are often sparse and the underlying mechanisms governing the biology are

gov.au/. MJS recieved funding from the Australian Research Council (DP200100177) https://www. arc.gov.au/. The funders had no role in study design, data collection and analysis, decision to publish, or preparation of the manuscript.

complex, such as those governing cell migration and proliferation [1, 2]. In addition to being sparse, the data are typically noisy [3]. These issues together make it difficult to understand the laws governing the biological process of interest as the sparsity of the data gives little information for establishing these laws, and the noise embedded in the data implies that any estimates are likely to involve significant uncertainty. In this work, we propose a novel framework for efficiently discovering the functional forms of various biological processes within an experiment from sparse noisy data, requiring far less computational effort for obtaining valid inference and associated uncertainty estimates than other methods in the literature for the same purpose. Our method naturally quantifies uncertainty in these learned functional forms through confidence intervals. We illustrate our approach using data from a commonly used scratch assay experiment which explores the dependence of an initial density of cells on the migration rate and proliferation rate of cells.

Identifying appropriate terms in a mathematical model is traditionally accomplished through empirical reasoning. Model discovery methods instead consider these terms as unknown and try to estimate them, typically writing partial differential equation (PDE) models in the form $u_t = \mathcal{N}(u, \partial_x, \partial_{xx}, \ldots, x, \boldsymbol{\theta})$ for some nonlinear function $\mathcal{N}$ to be learned and parameters $\boldsymbol{\theta}$ [4]. Some of the earliest methods for discovering the structure of a model were the symbolic regression methods of Bongard and Lipson [5] and Schmit and Lipson [6] which identify a model given some data, simultaneously weighting these found models by their complexity so that parsimonious models are obtained. Extensions to these methods that are robust for sparse data sets have recently been developed by the seminal work of Brunton et al. [7] and Rudy et al. [4] who consider an equation learning approach for model discovery for differential equations, using sparse regression methods to identify model terms from a large library of candidate terms. Sparse Bayesian learning methods have also been used [8]. Martina-Perez et al. [9] incorporate Bayesian methods into equation learning methods to quantify uncertainty in learned PDE models using agent-based simulations. More modern developments have been through neural networks. Raissi et al. [10] and Chen et al. [11] use physics informed neural networks to learn nonlinear PDEs from sparse data. More relevant to our work, Lagergren et al. [12] extended the concept of a physics informed neural network to that of a biologically informed neural network for equation learning and demonstrate their methods on experimental data from an experiment from Jin et al. [13]. This experiment describes the invasion of cells in a scratch assay, and Jin et al. [13] demonstrate how the cell migration rate depends on the initial cell density of the experiment.

Motivated by the goal of estimating the uncertainty of the learned equations, our work significantly extends these ideas by implementing an approach which replaces the neural network framework of Lagergren et al. [12] with a Gaussian process framework. Working with a Gaussian process allows us to develop a simple and efficient parametric bootstrapping approach to introduce uncertainty quantification with repeated sampling so that sampling distributions can be obtained, treating the Gaussian process as a surrogate for the underlying dynamical system governing the motion and proliferation of the population of cells. Gaussian processes have been successfully used by others for learning differential equations. Wang et al. [14] use Gaussian processes to learn ordinary differential equations (ODEs). Chen et al. [15, 16] use Gaussian processes to learn the functional forms for the coefficients in nonlinear PDEs. Other approaches using Gaussian processes are given in [17–20]. Our contribution differs from these previous approaches in that we specifically consider sparse data, in particular we consider data from a biological experiment rather than from synthetic data, meaning that we work with a relatively small number of data points, and we enable uncertainty quantification for learned results. We illustrate the work using data from Jin et al. [13], which describes the invasion of prostate cancer cells and their density $u$ as a function of space $x$ and time $t$.

Our framework can be thought of as an alternative to Bayesian methods for parameter estimation, uncertainty quantification, and model selection, such as those developed by Lambert et al. [21] for a spatial agent-based model. In these approaches, a statistical model is introduced and expressed in terms of the solution of some differential equations that depends on certain unknown parameters, and inference is performed through a defined likelihood function, using methods such as those reviewed by Hines et al. [22]. Bayesian approaches for parameter estimation are well-established [23–25], typically relying on sampling techniques such as Markov chain Monte Carlo (MCMC) [26], which can be computationally expensive, especially for PDE models [27]. Some features of our approach are advantageous relative to Bayesian approaches. For example, MCMC-based Bayesian approaches can be challenging because of the choice of hyperparameters, convergence properties, and the serial nature of these algorithms. In contrast, our approach is more naturally parallelisable, and it allows us to incorporate a candidate library of terms in the PDE model similar to the equation learning approach of Brunton and colleagues [4, 7]. Our approach is computationally efficient since we require a relatively small number of bootstrap samples for inference and uncertainty quantification whereas Bayesian methods can require many thousands of samples for the same purpose. In summary, our approach combines attractive features of both Bayesian model selection and equation learning, while avoiding some of the challenging features of both these previous approaches. For example, our framework naturally incorporates uncertainty quantification whereas standard applications of equation learning do not [4, 7].

Frequentist approaches can also be used for model selection and uncertainty quantification. For nested models, the likelihood ratio test is useful for performing model selection [28]. Information-criterion based approaches to model selection are also useful, allowing for comparison of non-nested models [29], and in this work we demonstrate how we can incorporate an information-criterion based approach to model selection with bootstrapping.

The structure of this manuscript is as follows. In Section 2 we demonstrate our results for several models fit to the data by Jin et al. [13], followed by a discussion of these results and recommendations for future work. Section 3 considers our methods in greater detail, outlining how we use Gaussian processes to learn the equations and details about optimisation and bootstrapping. Additional details are presented in S1 Text in S1 Document and S2 Text in S1 Document, and code for reproducing the work is available with our JULIA package at https://github.com/DanielVandH/EquationLearning.jl.

## 2 Results and discussion

In this section we discuss data reported by Jin et al. [13] and describe the fitted Gaussian processes. We then discuss the results for a set of models fit to these data and our model selection process, along with a discussion of the quantified uncertainty. We finish with a conclusion around these results and recommendations for future work.

Let us start by commenting briefly on the history of this data, and upon the broader questions that can be addressed using this data. The application of reaction-diffusion models to collective cell migration and proliferation dates back to 1990 with the work of Sherratt and Murray [30] who use solutions of the Fisher-Kolmogorov and Porous-Fisher models to interpret the shrinking radius of an epidermal wound (these models are introduced in Section 2.3). This important study motivated the question of whether collective cell migration is best described by a linear diffusion mechanism as in the Fisher-Kolmogorov model, or a nonlinear degenerate diffusion term as in the Porous-Fisher model [31]. Comparing experimental measurements to the solutions of these PDE models, Sherratt and Murray [30] were unable to draw definitive conclusions about whether linear or nonlinear diffusion was the most

appropriate model for their experimental data. The more recent work by Jin et al. [13] led to similar conclusions since they found that both the Fisher-Kolmgorov model with linear diffusion provided just as good a match to their data as the Porous-Fisher model with nonlinear diffusion for a new set of experimental data describing a wound scratch assay using prostate cancer cells. More recently, Lagergren et al. [12] re-examined the experimental data reported by Jin et al. [13] and use a novel equation learning-based approach to discover the terms in a reaction-diffusion model such that the solution of that model provided the best match to the experimental data. However, the approach used by Lagergren et al. [12] did not incorporate any uncertainty quantification so it is difficult to reliably distinguish between the suitability of linear and nonlinear diffusion mechanisms using this approach. A key aim of the present study is to extend the analysis of Lagergren et al. [12] by using a Gaussian process together with bootstrapping to reveal the mechanisms in a reaction-diffusion framework that best explain the data reported by Jin et al. [13] in an attempt to distinguish between the roles of linear and nonlinear diffusion with uncertainty quantification rather than just relying on point estimates of parameters only.

## 2.1 Scratch assay data

The data we consider in this manuscript is a monolayer scratch assay experiment from Jin et al. [13] using the ZOOM™ system. In this experiment, a cell monolayer is first grown up to some initial cell density; the cells come from the PC-3 prostate cancer cell line [32]. Cells are then placed at various densities in 96-well ImageLock plates (Essen BioScience), with cells uniformly distributed in each well at pre-specified initial cell densities, which we refer to as the *initial number of cells per well*. Following seeding, a scratch is created, using a WoundMaker™, in the middle of the cell monolayer to create a vacant region into which the remaining cells invade, as shown in Fig 1(a) for a single well. Since the cells are distributed evenly, and as they remain spatially uniform away from the edges, the analysis of the cells focuses wholly on the cells in the blue rectangle shown in Fig 1(a). Images of the experiment taken at equally spaced time intervals are shown at $t = 0$, 12, 24, 36, and 48 h. Experiments are prepared with 10,000, 12,000, 14,000, 16,000, 18,000, and 20,000 initial cells per well (CPW) and some initial cell distributions for these densities are shown in Fig 1(b)–1(g). The initially vacant region in the middle of the monolayer is visually obvious immediately after scratching. For each initial cell density, three identically prepared experiments are obtained, giving three data sets for each initial cell density, as shown in Fig 1(h) which shows the evolution of three experiments over the two day span of the experiment, where we see that there is some variability across each experiment and the vacant region in each case is filled by the end of the experiment.

Fig 2(a) and 2(b) show two images from the experiment of Jin et al. [13] at 12,000 and 20,000 CPW, respectively, at the shown times. These cell positions take up two-dimensional space, and thus to represent the cell invasion with a one-dimensional profile we use cell densities by counting the number of cells as follows. First, the images are divided into equally spaced columns centred at $x = 25, 75, \ldots, 1925$ µm, shown in blue in Fig 2(c). The number of cells per column are then counted using Adobe Photoshop. This number of cells is then divided by the area of the column, giving an estimate for the cell density $u$ at each point $(x, t)$. The carrying capacity density $K$ is measured directly by taking the number of cells at $t = 48$ h, when the cells are most confluent, as shown in Fig 2(b), for the 20,000 CPW data and estimating the cell density in the blue regions in Fig 2(d), giving an estimate $K = 1.7 \times 10^{-3}$ cells µm$^{-2}$. We note that the carrying capacity density, $K$, is the only quantity that can be directly measured from these experiments, thus we treat $K$ as a fixed quantity, setting $K = 1.7 \times 10^{-3}$ cells µm$^{-2}$. The remaining quantities in this study are all derived using our mechanism learning framework.

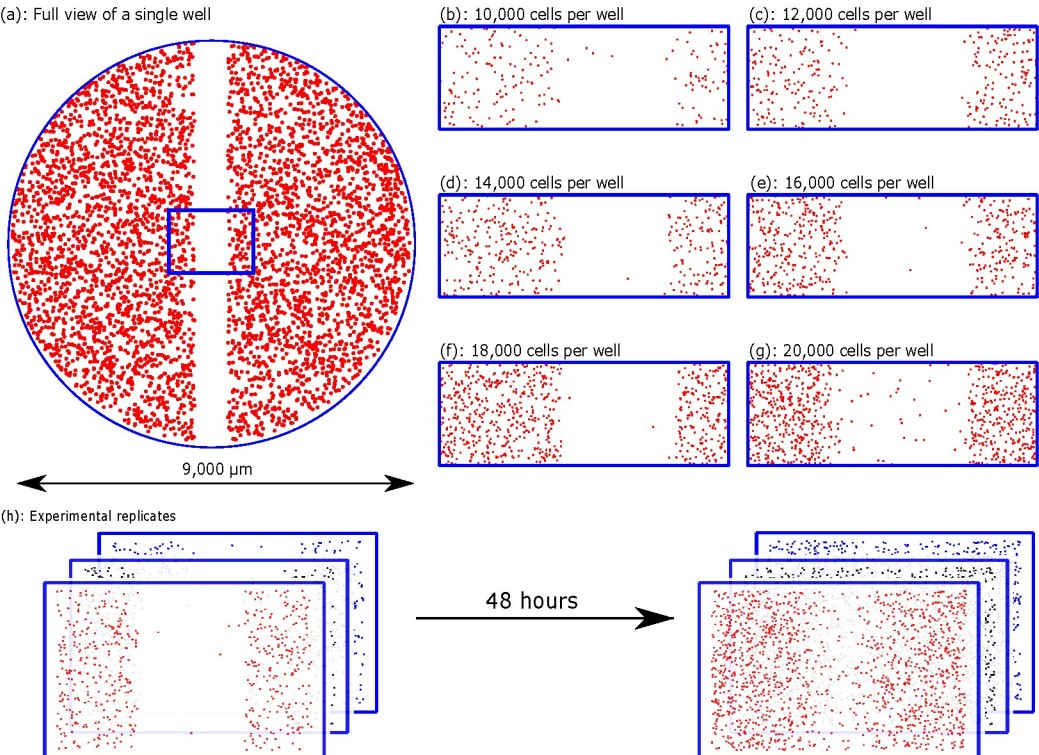

**Fig 1. Scratch assay data.** Schematics of the scratch assay experiments performed by Jin et al. [13]. (a) A view of a single well. The 1900 µm wide blue rectangle shows the region where data are imaged and analysed. (b)–(g) Example initial distribution of cells for each initial number of CPW corresponding to the title of each plot. (h) Growth of three initial cell distributions for a single initial CPW, at $t = 0$ h (left) and $t = 48$ h (right).

We summarise all data from Jin et al. [13] at each initial cell density in Fig 3. For the lower cell densities, notice that the cells take longer to invade into the middle of the wells where the scratch originates, with most data sets never meeting the carrying capacity density. In contrast, Fig 3(f) shows that the 20,000 CPW data quickly invade this interior region, essentially at the carrying capacity density at $t = 48$ h. For the data itself, the average densities at each time are shown using the continuous curves. We see that the data are reasonably noisy around these lines; methods for de-noising this data are not considered in this work. Jin et al. [13] use these data and least squares estimation to conclude that these experiments, and their reproducibility, is highly dependent on the initial cell density, which is often overlooked in the literature. We extend the work of Jin et al. [13] and consider more models, verifying their results and providing uncertainty quantification to the parameter estimates that they obtain.

## 2.2 Gaussian processes

Gaussian processes are a core component of our approach, providing us with a means of obtaining sampling distributions for parameters, functional forms, cell densities, and model rankings. Our procedure starts by fitting a Gaussian process to our spatiotemporal data. Gaussian processes are a nonparametric generalisation of a Gaussian probability distribution, instead defining a distribution over all functions, living in an infinite-dimensional space [33]. For our cell density data, we fit a Gaussian process to each data set and show the results in Fig 4. These curves are shown using the solid lines, and the confidence intervals around these

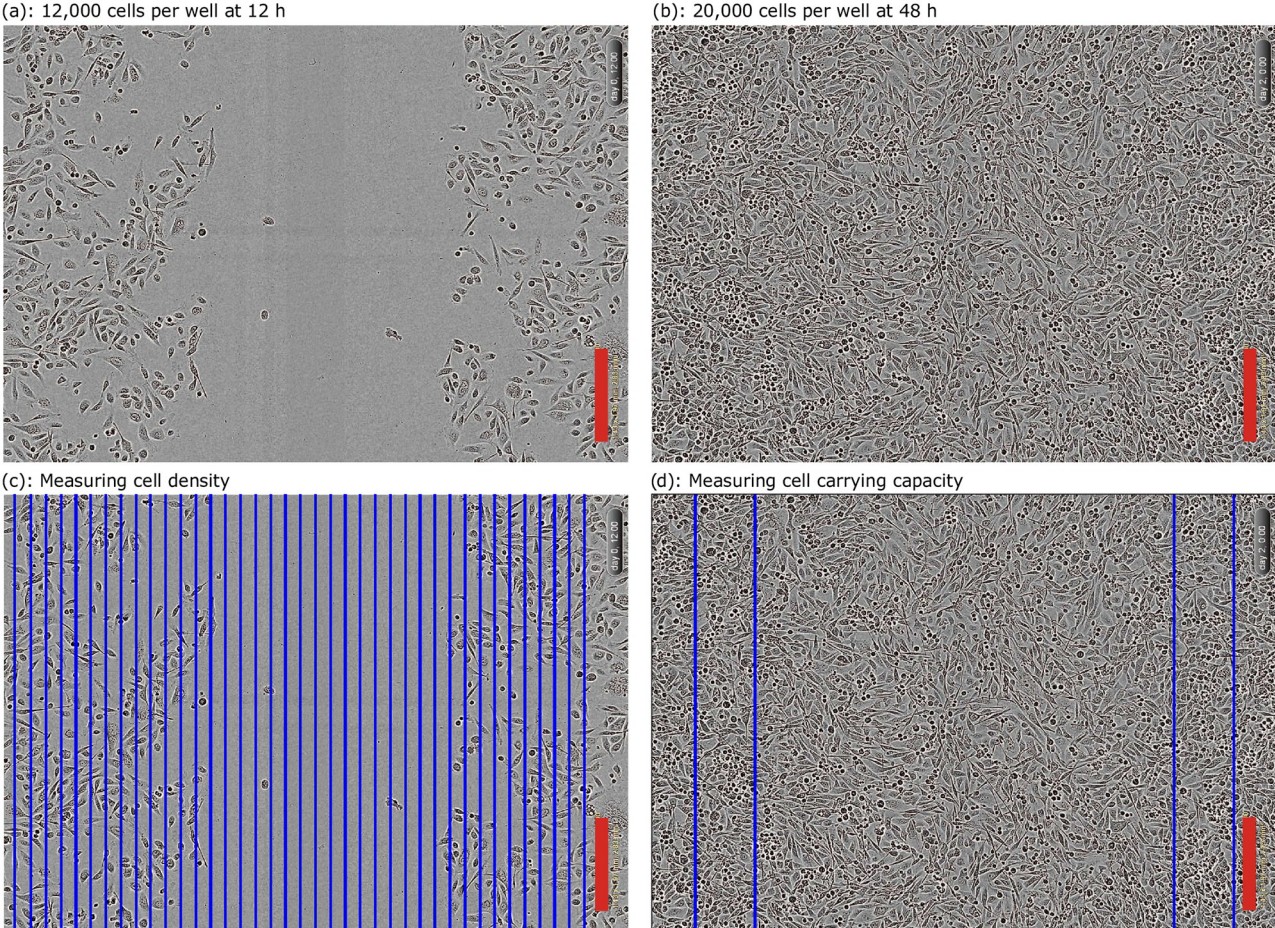

**Fig 2. Measuring cell density cell carrying capacity.** (a) and (b) are two images from the experiment of Jin et al. [13] with the initial CPW and times shown. The red scale bar corresponds to 300 μm. (c) Illustration of how the images are split into columns, shown in blue, with cells counted for each column to measure the cell densities. (d) The carrying capacity density is measured by counting the number of cells for the 20,000 CPW data set at $t$ = 48 h in the blue rectangles shown, divided by the area of these regions.

curves are shown with the transparent ribbons. We note that these confidence intervals do not precisely capture the data points, and this imprecision is intended as we are trying to capture the underlying function governing the cell densities rather than the noisy cell densities themselves. If we were to consider capturing the data itself, an extra term could be added to the variance estimates [33].

An alternative view for these Gaussian processes is shown in Fig 5 where we show a space-time diagram for the fitted models. These figures show the evolution of cells in the $(x, t)$-plane, with the colour corresponding to the cell density. It is to this grid of $(x, t)$ values that our Gaussian processes are defined over. The figures make it clearer that the interior regions have far fewer cells than around the boundaries of the wells, and that the 20,000 CPW data in Fig 5 (f) has almost reached the carrying capacity density for later times.

The primary motivation for using Gaussian processes is that they allow us to obtain samples that match the data, treating the Gaussian process as a surrogate for the dynamical system governing the data-generating process, i.e. the system without the experimental noise. Gaussian processes are attractive relative to other methods as they are smooth, differentiable, and easy to

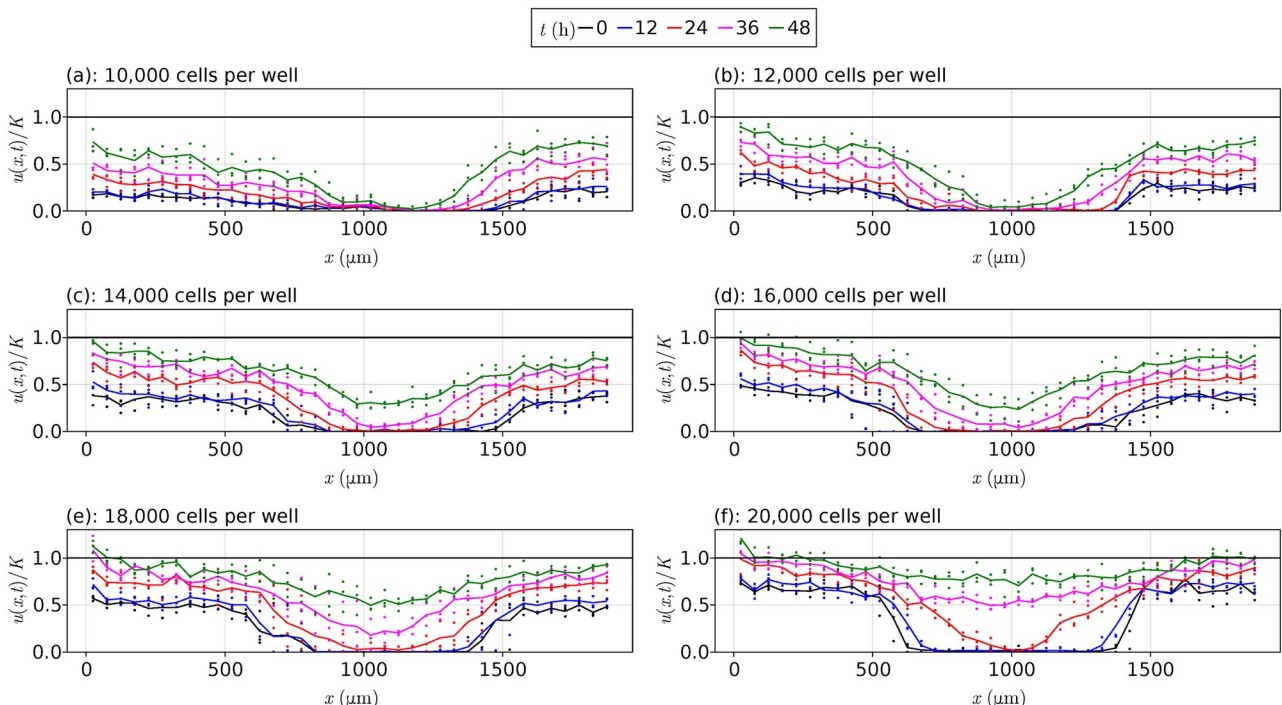

**Fig 3. Scratch assay data.** Scratch assay data from Jin et al. [13] for each initial cell density, with the vertical axis given by $u/K$ rather than $u$, where $K = 1.7 \times 10^{-3}$ cells $\mu m^{-2}$. In each plot, the points represent experimental observations, and the continuous lines represent averaged values at each time over the three replications. The black horizontal lines are at $u/K = 1$.

sample from. These properties are important when working with a PDE model where we are interested in estimating $u(x, t)$, $\partial u(x, t)/\partial t$, $\partial u(x, t)/\partial x$ and $\partial^2 u(x, t)/\partial x^2$, even in the presence of noise. The Gaussian process samples for the function values and derivatives lead to an optimisation problem that determines the relevant mechanisms. This optimisation problem, outlined in Section 3, uses a loss function that jointly constrains the Gaussian process to match the proposed PDE, using the sampled function values and their derivatives, and the solutions to a PDE with the proposed parameters to match the experimental data. By applying a parametric bootstrapping procedure and repeatedly sampling from the Gaussian process, we replicate repeated collection of non-noisy data from the experiments. The data sets from each bootstrap realisation enable us to fit multiple proposed models, giving individual sets of parameters for each model, and allow us to compare models across various realisations. Hence, obtaining a complete set of bootstrap realisations, the Gaussian process enables us to obtain sampling distributions for parameters from each model and also a sampling distribution for the rankings of the candidate models. These ideas are made clear in the following sections, and their implementation is described in Section 3. We note that we use the Gaussian process as a surrogate for the dynamical system governing the data-generating process rather than the data-generating process itself, and this is the reason that an approach like bootstrapping the residuals of a simpler non-parametric model, following for example the discussion by Fieberg et al. [34], could not be used, hence we choose to work with a Gaussian process. Moreover, while the uncertainty quantification and inference we obtain is focused on the underlying dynamical system rather than the data-generating process, the uncertainty can be interpreted relative to the data-generating process. In particular, the noise from the experimental data is

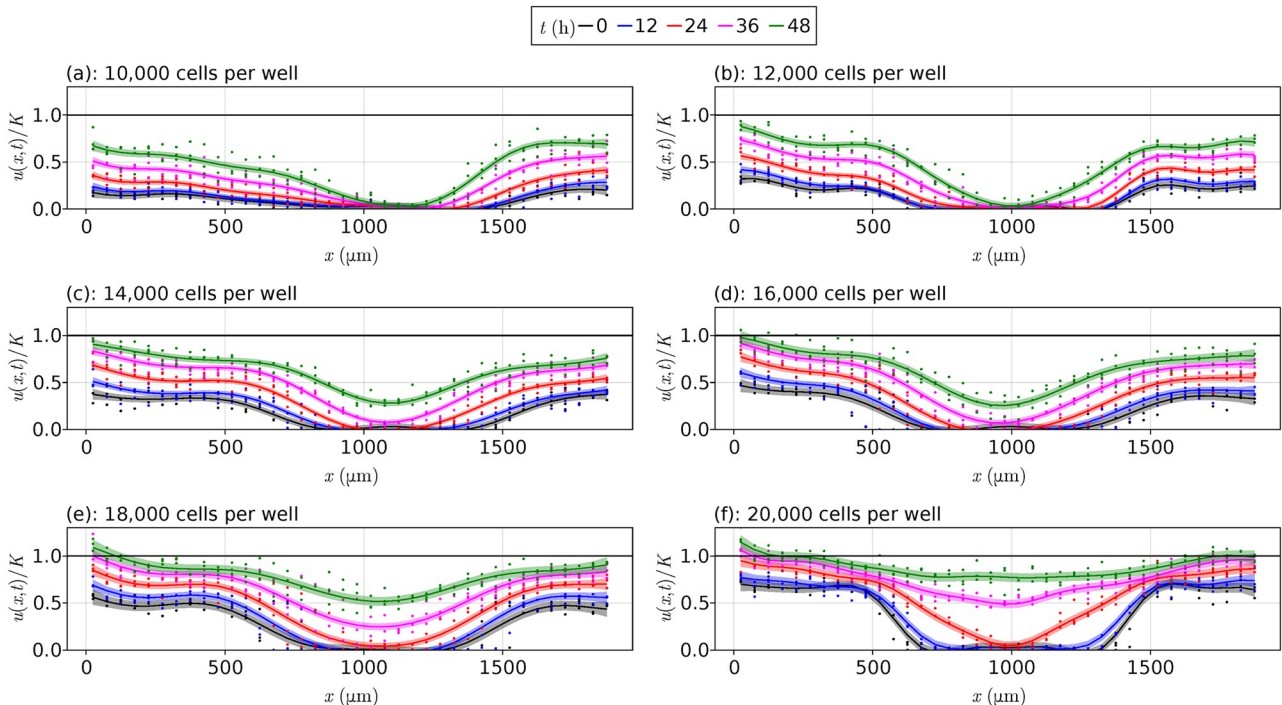

**Fig 4. Fitted Gaussian processes.** Fitted Gaussian processes to the data from Jin et al. [13] for each initial cell density, with the vertical axis given by $u/K$ rather than $u$, where $K = 1.7 \times 10^{-3}$ cells $\mu m^{-2}$. The points are the experimental data, the smooth curves are the fitted Gaussian process curves, and the shaded region represents the uncertainty around these curves implied by the Gaussian process. We estimate the uncertainty in these estimates by shading the interval corresponding to the mean plus or minus the standard deviation. The black horizontal lines are at $u/K = 1$.

propagated into our estimates for the de-noised dynamical system, thus any variability in our results are due to the experimental noise.

By treating the Gaussian process as a surrogate for the dynamical system governing the data-generating process, we implicitly rely on the suitability of the Gaussian process as an appropriate surrogate when making inference and providing uncertainty quantification. For instance, if the confidence intervals in Fig 4 were too large relative to the fluctuations in the data itself, any inferences made from the Gaussian process samples would be irrelevant as the Gaussian process would not necessarily be a reasonable model of the data-generating process. Moreover, if the density profiles in Fig 4 were too smooth, the results obtained could be undesirably smooth relative to the data and inference could be difficult to interpret. Hence, there is a trade-off between the smoothness of the Gaussian process and the uncertainty around each point of the Gaussian process, and both factors are crucial for the uncertainty quantification in our results and any inferences made.

## 2.3 Mechanism discovery

Let us now consider the problem of discovering the mechanisms driving the invasion process in the data by Jin et al. [13]. For the purpose of this manuscript we consider PDE models that take the form

$$\frac{\partial u}{\partial t} = \underbrace{T(t; \boldsymbol{\alpha})}_{\text{delay}} \left[ \overbrace{\frac{\partial}{\partial x} \left( \underbrace{D(u; \boldsymbol{\beta})}_{\text{nonlinear diffusivity}} \frac{\partial u}{\partial x} \right)}^{\text{cell migration}} + \overbrace{R(u; \boldsymbol{\gamma})}^{\text{cell proliferation}} \right], \tag{1}$$

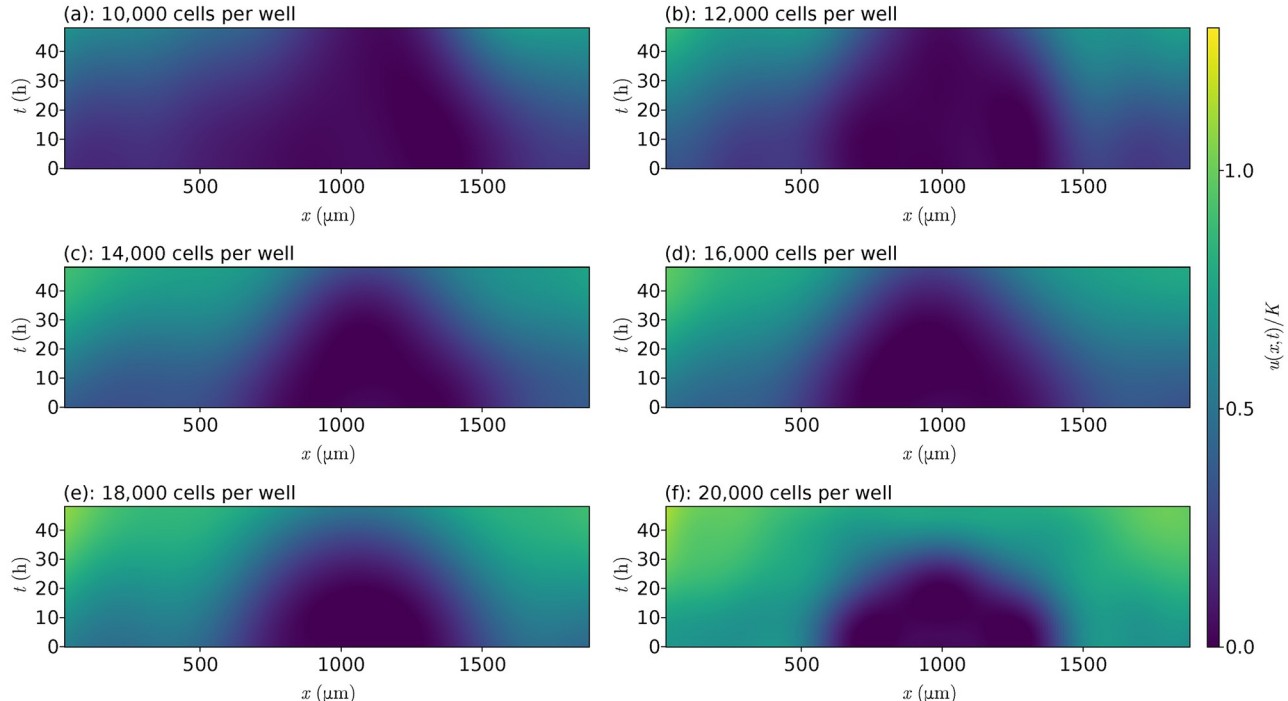

**Fig 5. Space-time diagrams.** Space-time diagrams from the fitted Gaussian processes to the data from Jin et al. [13] for each initial cell density, coloured according to $u/K$ where $K = 1.7 \times 10^{-3}$ cells μm$^{-2}$.

where $u = u(x, t)$ is the cell density at position $x$ and time $t$. We assume that there are three mechanisms driving the invasion: a delay term $T(t; \boldsymbol{\alpha})$ that scales migration and proliferation equally, a nonlinear diffusivity term $D(u; \boldsymbol{\beta})$ describing cell migration, and a reaction term $R(u; \gamma)$ describing cell proliferation. This model (1) is the same as that considered by Lagergren et al. [12]. The parameters $\boldsymbol{\alpha}, \boldsymbol{\beta}$, and $\gamma$ for delay, diffusion, and reaction, respectively, are estimated based on an assumed form for the functions. We note that we are using knowledge of cell biology to inform us about what the general class of models may look like. If we used for example the candidate library methods of Brunton et al. [7] and Rudy et al. [4], we might find a similarly structured model but may not for example satisfy a conservation form as our model does.

For this study we consider $T(t; \boldsymbol{\alpha}) = 1$, meaning no delay, or $T(t; \boldsymbol{\alpha}) = 1/[1 + \exp(-\alpha_1 - \alpha_2 t)]$, meaning that logistic delay is present [12]. Incorporating the possibility of having a delay is motivated by separate experimental measurements by Jin et al. [35] who showed that the physical act of scratching the monolayer of cells can lead to a temporary delay in normal cell function. We also assume that the reaction term $R(u; \gamma)$ takes the logistic form $R(u; \gamma) = \gamma_1 u(1 - u/K)$, but our methods apply equally well to other forms, such as Gompertz growth models [36]. Previous analysis of this data suggests that cell migration is the most uncertain mechanism [13]. Therefore, we pay particular attention to the nonlinear diffusivity term and consider several different forms for $D(u; \boldsymbol{\beta})$. The simplest, most commonly used approach is to treat $D(u; \boldsymbol{\beta})$ as a constant $D(u; \boldsymbol{\beta}) = \beta_1$, as is the case in the well-studied Fisher-Kolmogorov model [37]. An alternative model for the nonlinear diffusivity function is $D(u; \boldsymbol{\beta}) = \beta_2 u/K$, which assumes that cell migration is an increasing function of density, as in the Porous-Fisher model [30]. The final nonlinear diffusion model we consider is $D(u; \boldsymbol{\beta}) = \beta_1 + \beta_2 (u/K)^{\beta_3}$;

this model is a hybrid of the Fisher-Kolmogorov and Porous-Fisher models with an exponent on $\beta_3$ to be estimated. This hybrid model was recently introduced by Lagergren et al. [12]. With these terms, we consider the following five combinations of these mechanisms:

$$\text{Model 1}: T(t; \boldsymbol{\alpha}) = 1, \qquad\qquad D(u; \boldsymbol{\beta}) = \beta_1, \tag{2}$$

$$\text{Model 2}: T(t; \boldsymbol{\alpha}) = \frac{1}{1 + \exp(-\alpha_1 - \alpha_2 t)}, \; D(u; \boldsymbol{\beta}) = \beta_1, \tag{3}$$

$$\text{Model 3}: T(t; \boldsymbol{\alpha}) = 1, \qquad\qquad D(u; \boldsymbol{\beta}) = \beta_2\left(\frac{u}{K}\right), \tag{4}$$

$$\text{Model 4}: T(t; \boldsymbol{\alpha}) = \frac{1}{1 + \exp(-\alpha_1 - \alpha_2 t)}, \; D(u; \boldsymbol{\beta}) = \beta_2\left(\frac{u}{K}\right), \tag{5}$$

$$\text{Model 5}: T(t; \boldsymbol{\alpha}) = \frac{1}{1 + \exp(-\alpha_1 - \alpha_2 t)}, \; D(u; \boldsymbol{\beta}) = \beta_1 + \beta_2\left(\frac{u}{K}\right)^{\beta_3}, \tag{6}$$

using the same reaction mechanism $R(u; \boldsymbol{\gamma}) = \gamma_1 u(1 - u/K)$ for each model. We fit these models to each data set and then compare them using the Akaike information criterion (AIC) [38] as described in Section 3. To summarise the procedure used for fitting these models, we start by fitting a Gaussian process to the observed data, computing the mean vector and covariance matrix for the joint distribution of the data and the spatial and temporal derivatives of the density. For a given bootstrap sample, we sample the density values and its derivatives from this Gaussian process. The data from these samples are used to solve the corresponding PDE model, whose initial condition we sample from the Gaussian process or take directly from the data, allowing parameters to be estimated by minimising the errors between the left-hand and right-hand sides of the proposed PDE, and also between the numerical solution of the PDE and the experimental data. This process is repeated $B$ times to give a sample $(\hat{\boldsymbol{\theta}}^{(1)}, \ldots, \hat{\boldsymbol{\theta}}^{(B)})$ of parameter values for the model; for example, for (2) this sample is made up of the samples $(\beta_1^{(1)}, \ldots, \beta_1^{(B)})$ and $(\gamma_1^{(1)}, \ldots, \gamma_1^{(B)})$ for the diffusivity and proliferation rates, respectively. Section 3 gives a more thorough discussion of how these methods are implemented.

**2.3.1 Preliminary synthetic data analysis.** To demonstrate the veracity of our methods, we consider some synthetic data analysis (see S1 Text in S1 Document). In these studies we simulate data according to some prescribed model, using the 10,000 CPW data from Jin et al. [13] to define the initial condition for the PDEs. In the first synthetic data study, we consider the model

$$\frac{\partial u}{\partial t} = \beta_1 \frac{\partial^2 u}{\partial x^2} + \gamma_1 u\left(1 - \frac{u}{K}\right), \tag{7}$$

where $\beta_1 = 301 \; \mu\text{m}^2 \, \text{h}^{-1}$ and $\gamma_1 = 0.044 \; \text{h}^{-1}$. We learn from this study that we can successfully capture these parameter estimates within their respective confidence intervals. The second study considers more models, fitting all five of the models in (2)–(6), with the true model given by

$$\frac{\partial u}{\partial t} = \frac{1}{1 + \exp(-\alpha_1 - \alpha_2 t)}\left[\beta_1 \frac{\partial^2 u}{\partial x^2} + \gamma_1 u\left(1 - \frac{u}{K}\right)\right], \tag{8}$$

where $\alpha_1 = -1.5$, $\alpha_2 = 0.31 \; \text{h}^{-1}$, $\beta_1 = 571 \; \mu\text{m}^2 \, \text{h}^{-1}$, and $\gamma_1 = 0.081 \; \text{h}^{-1}$. We again learn from this study that we can successfully re-estimate our model, with reasonable confidence intervals for

all the parameters and PDE solutions, and our procedure for model selection identifies the correct model with high probability. We also demonstrate how model misspecification may present itself. In particular, a missing delay term can be identified by observing if the learned PDE solutions initially underestimate (or overestimate) the data, and then as time progresses the solution then overestimates (or underestimates) the data. If the diffusion term is misspecified we can compare the solution of the learned PDE with the data. If the match is poor it is likely that the diffusion term is misspecified. Our approach is always to work with the simplest possible models in the first instance, before working with more complicated terms in the governing equations, since this approach can avoid working with overly complex models that are difficult to interpret. Further details and discussion are given in S1 Text in S1 Document.

We also consider in S2 Text in S1 Document an efficient procedure for problems with no delay and whose nonlinear diffusivity and reaction functions can be written as a linear combination of the parameters to be estimated, which we refer to as the *basis function approach*. We apply this method to the same simulation study that considered (7) and show that we get the same results, as we show in S2 Fig I in S1 Document, and the computation time is significantly decreased. However, this approach may not be suitable for providing final results as it does not directly consider the data and thus leads to greater PDE errors than with the approach presented here. The approach can be useful for obtaining initial scale estimates for the parameters quickly for linear problems that can help improve the optimisation procedure using the main approach. We also note that this basic function approach could be applied to nonlinear models, like those that we consider in Section 3, by iteratively fixing parameters and optimising the others in a way that makes the problem linear, thus providing a quick way for obtaining scales or results for problems in general. See S2 Text in S1 Document for more detail. The rest of this manuscript only shows results from the more expensive, and far more general, method described in Section 3.

**2.3.2 Experimental data.** We now apply our methods to the data from Jin et al. [13]. For the purpose of fitting these models, we follow Lagergren et al. [12] by excluding the left-most density measurement for each data set at $x = 25$ μm as this is consistently larger than the other density measurements, as shown in Fig 3. This removal gives us $N = 37$ points in space, $M = 5$ points in time, and $NM = 185$ total points per experimental replicate. We consider fitting each of the models (2)–(6) to each data set. To quantify the error between the solutions and the data, we estimate the median absolute percentage errors (MAPE)

$$\mathcal{E}^{(b)} = \text{median}\{e_{ij}^{(b)}\}, \quad \text{where} \quad e_{ij}^{(b)} = 100 \frac{|\hat{u}_{ij}^{(b)} - u_{ij}|}{\max\{\zeta, u_{ij}\}}, \tag{9}$$

where $i = 1, \ldots, N$ and $j = 1, \ldots, M$ range over the number of spatial gridpoints and the number of temporal gridpoints, respectively. Here, the superscript $b$ refers to the results from the $b$th set of results from our sampled Gaussian process. We use $u_{ij}$ to denote the average of the replicates for the cell densities at the point $x_i$ and time $t_j$. The term $\hat{u}_{ij}^{(b)}$ is the numerical approximation for this density that comes from the numerical solution of the corresponding PDE model. We set $\zeta$ to be the machine precision, $\zeta \approx 2.2 \times 10^{-16}$. We then take quantiles for the sample of MAPEs $\{\mathcal{E}^{(1)}, \ldots, \mathcal{E}^{(B)}\}$, where $B$ is the number of bootstrap replicates, at the 2.5% and 97.5% levels to give an approximate 95% confidence interval for the MAPE. Remaining details, such as for parameter estimation and bootstrapping, are given in Section 3.

Table 1 summarises the estimated parameters in (2)–(6) for each data set. For each model, we see that the Fisher-Kolmogorov model with delay, Eq (3), is the best fitting model, as shown by the blue columns. We measure this goodness of fit using the probabilities $\mathbb{P}(E_1)$, $\mathbb{P}(E_2)$, and $\mathbb{P}(E_3)$, which give the probability that the model is in the class of best fitting models,

**Table 1. Model selection results for the five models considered.**

| | Models | | | | |
|---|---|---|---|---|---|
| | **Model 1** | **Model 2** | **Model 3** | **Model 4** | **Model 5** |
| **10,000 cells per well** | | | | | |
| PDE Error (%) | (20.864, 32.123) | (13.943, 23.048) | (21.158, 32.887) | (14.554, 23.934) | (14.219, 23.564) |
| $\alpha_1$ | — | (−3.456, −0.217) [−1.464] | — | (−3.863, −0.372) [−1.514] | (−2.688, −0.637) [−1.118] |
| $\alpha_2$ (h$^{-1}$) | — | (0.091, 0.373) [0.240] | — | (0.084, 0.402) [0.174] | (0.104, 0.341) [0.166] |
| $\beta_1$ (μm$^2$ h$^{-1}$) | (54.396, 289.822) [164.829] | (51.93, 365.923) [203.149] | — | — | (92.394, 326.322) [99.486] |
| $\beta_2$ (μm$^2$ h$^{-1}$) | — | — | (20.608, 979.03) [429.285] | (13.828, 849.614) [307.338] | (−254.821, 1065.03) [−153.593] |
| $\beta_3$ | — | — | — | — | (1.927, 3.286) [3.018] |
| $\gamma_1$ (h$^{-1}$) | (0.045, 0.052) [0.049] | (0.054, 0.063) [0.058] | (0.045, 0.052) [0.049] | (0.053, 0.065) [0.057] | (0.053, 0.062) [0.058] |
| $\mathbb{P}(E_1)$ | 0.00 | 0.96 | 0.0 | 0.07 | 0.04 |
| $\mathbb{P}(E_2)$ | 0.01 | 0.00 | 0.0 | 0.03 | 0.23 |
| $\mathbb{P}(E_3)$ | 0.99 | 0.04 | 1.0 | 0.90 | 0.73 |
| **12,000 cells per well** | | | | | |
| PDE Error (%) | (15.736, 21.384) | (10.104, 14.547) | (15.23, 21.331) | (9.747, 14.411) | (10.282, 14.783) |
| $\alpha_1$ | — | (−3.088, −0.854) [−1.619] | — | (−3.055, −0.826) [−1.574] | (−2.403, −1.092) [−1.847] |
| $\alpha_2$ (h$^{-1}$) | — | (0.133, 0.307) [0.202] | — | (0.13, 0.303) [0.189] | (0.16, 0.253) [0.220] |
| $\beta_1$ (μm$^2$ h$^{-1}$) | (91.735, 193.201) [141.416] | (117.181, 222.334) [170.212] | — | — | (106.845, 219.685) [169.49] |
| $\beta_2$ (μm$^2$ h$^{-1}$) | — | — | (207.898, 584.976) [423.608] | (194.568, 531.537) [320.266] | (−437.945, −179.991) [−328.306] |
| $\beta_3$ | — | — | — | — | (7.306, 9.143) [8.261] |
| $\gamma_1$ (h$^{-1}$) | (0.044, 0.049) [0.046] | (0.055, 0.062) [0.057] | (0.044, 0.048) [0.046] | (0.054, 0.061) [0.057] | (0.054, 0.062) [0.057] |
| $\mathbb{P}(E_1)$ | 0.00 | 1.00 | 0.00 | 0.00 | 0.16 |
| $\mathbb{P}(E_2)$ | 0.00 | 0.00 | 0.00 | 0.00 | 0.49 |
| $\mathbb{P}(E_3)$ | 1.00 | 0.00 | 1.00 | 1.00 | 0.35 |
| **14,000 cells per well** | | | | | |
| PDE Error (%) | (11.794, 14.811) | (10.147, 14.635) | (10.598, 16.571) | (9.996, 15.573) | (9.9, 17.135) |
| $\alpha_1$ | — | (−2.858, −0.866) [−1.091] | — | (−2.588, −0.744) [−0.873] | (−2.395, −1.12) [−1.573] |
| $\alpha_2$ (h$^{-1}$) | — | (0.199, 0.401) [0.229] | — | (0.19, 0.401) [0.233] | (0.232, 0.361) [0.269] |
| $\beta_1$ (μm$^2$ h$^{-1}$) | (423.353, 689.093) [559.121] | (452.419, 763.335) [624.576] | — | — | (339.505, 753.689) [648.980] |
| $\beta_2$ (μm$^2$ h$^{-1}$) | — | — | (956.516, 2020.74) [1576.910] | (841.97, 2012.11) [1370.458] | (−903.293, 803.306) [−560.316] |
| $\beta_3$ | — | — | — | — | (1.552, 3.66) [3.279] |
| $\gamma_1$ (h$^{-1}$) | (0.047, 0.052) [0.052] | (0.054, 0.061) [0.058] | (0.047, 0.052) [0.049] | (0.052, 0.059) [0.056] | (0.054, 0.061) [0.057] |
| $\mathbb{P}(E_1)$ | 0.05 | 0.84 | 0.04 | 0.01 | 0.20 |
| $\mathbb{P}(E_2)$ | 0.04 | 0.09 | 0.01 | 0.01 | 0.20 |
| $\mathbb{P}(E_3)$ | 0.91 | 0.07 | 0.95 | 0.98 | 0.60 |
| **16,000 cells per well** | | | | | |
| PDE Error (%) | (10.648, 14.259) | (8.407, 12.702) | (10.217, 15.381) | (8.876, 14.332) | (8.183, 13.768) |
| $\alpha_1$ | — | (−2.256, 0.022) [−1.024] | — | (−2.17, −0.1) [−1.203] | (−2.636, −0.449) [−2.034] |
| $\alpha_2$ (h$^{-1}$) | — | (0.097, 0.335) [0.180] | — | (0.102, 0.34) [0.213] | (0.218, 0.477) [0.363] |

(*Continued*)

**Table 1.** (Continued)

| Models | | | | | |
|---|---|---|---|---|---|
| | **Model 1** | **Model 2** | **Model 3** | **Model 4** | **Model 5** |
| $\beta_1$ (µm² h⁻¹) | (382.49, 647.908) [518.476] | (434.293, 727.624) [595.310] | — | — | (380.617, 732.718) [478.344] |
| $\beta_2$ (µm² h⁻¹) | — | — | (808.965, 1581.16) [1068.249] | (728.143, 1549.18) [1126.042] | (−623.709, 183.84) [−400.083] |
| $\beta_3$ | — | — | — | — | (4.582, 7.115) [5.833] |
| $\gamma_1$ (h⁻¹) | (0.049, 0.055) [0.052] | (0.056, 0.067) [0.060] | (0.048, 0.054) [0.051] | (0.055, 0.065) [0.059] | (0.052, 0.064) [0.059] |
| $\mathbb{P}(E_1)$ | 0.01 | 0.92 | 0.00 | 0.07 | 0.15 |
| $\mathbb{P}(E_2)$ | 0.00 | 0.01 | 0.00 | 0.00 | 0.22 |
| $\mathbb{P}(E_3)$ | 0.99 | 0.07 | 1.00 | 0.93 | 0.63 |
| **18,000 cells per well** | | | | | |
| PDE Error (%) | (7.949, 10.907) | (6.609, 8.923) | (8.772, 12.14) | (7.152, 9.74) | (6.663, 10.902) |
| $\alpha_1$ | — | (−1.45, 0.066) [−0.765] | — | (−1.436, −0.022) [−0.671] | (−1.173, −0.453) [−0.994] |
| $\alpha_2$ (h⁻¹) | — | (0.059, 0.236) [0.136] | — | (0.056, 0.207) [0.117] | (0.085, 0.194) [0.140] |
| $\beta_1$ (µm² h⁻¹) | (433.393, 760.833) [565.126] | (470.944, 922.542) [654.164] | — | — | (300.168, 950.728) [653.921] |
| $\beta_2$ (µm² h⁻¹) | — | — | (655.963, 1614.74) [1016.562] | (561.39, 1724.61) [1085.503] | (−579.707, 177.486) [−408.079] |
| $\beta_3$ | — | — | — | — | (4.54, 6.519) [5.066] |
| $\gamma_1$ (h⁻¹) | (0.054, 0.061) [0.059] | (0.064, 0.088) [0.070] | (0.053, 0.06) [0.057] | (0.064, 0.09) [0.070] | (0.065, 0.087) [0.069] |
| $\mathbb{P}(E_1)$ | 0.04 | 0.76 | 0.00 | 0.00 | 0.32 |
| $\mathbb{P}(E_2)$ | 0.01 | 0.11 | 0.00 | 0.00 | 0.07 |
| $\mathbb{P}(E_3)$ | 0.95 | 0.13 | 1.00 | 1.00 | 0.61 |
| **20,000 cells per well** | | | | | |
| PDE Error (%) | (8.76, 11.287) | (7.206, 9.61) | (10.896, 14.895) | (8.211, 11.926) | (7.293, 10.536) |
| $\alpha_1$ | — | (−4.544, −1.961) [−2.434] | — | (−2.402, −1.721) [−2.068] | (−1.713, −0.891) [−1.014] |
| $\alpha_2$ (h⁻¹) | — | (0.301, 0.555) [0.340] | — | (0.269, 0.384) [0.320] | (0.15, 0.24) [0.215] |
| $\beta_1$ (µm² h⁻¹) | (401.865, 664.394) [542.045] | (443.875, 852.295) [655.544] | — | — | (506.936, 1251.44) [863.584] |
| $\beta_2$ (µm² h⁻¹) | — | — | (628.934, 1170.63) [880.493] | (627.656, 1461.9) [850.253] | (−1080.17, −433.677) [−747.381] |
| $\beta_3$ | — | — | — | — | (1.142, 1.477) [1.316] |
| $\gamma_1$ (h⁻¹) | (0.069, 0.077) [0.072] | (0.083, 0.093) [0.088] | (0.072, 0.084) [0.079] | (0.086, 0.099) [0.094] | (0.082, 0.099) [0.085] |
| $\mathbb{P}(E_1)$ | 0.00 | 0.84 | 0.00 | 0.00 | 0.26 |
| $\mathbb{P}(E_2)$ | 0.00 | 0.09 | 0.00 | 0.00 | 0.09 |
| $\mathbb{P}(E_3)$ | 1.00 | 0.07 | 1.00 | 1.00 | 0.65 |

Interval estimates, PDE errors, and model selection results for the models in (3)–(6) when applied to the each data set from Jin et al. [13]. The numbers in square brackets give the modes from the kernel density estimates of the corresponding bootstrap sample for the corresponding parameter. The PDE errors are computed using sampled initial conditions. Significance levels for all confidence intervals are 95%. The blue column for each data set highlights the optimal model, and the red text indicates confidence intervals that contain 0.

that there is only some evidence that the model is in this class, and that there is no evidence, respectively; these probabilities are discussed in Section 3, and should be interpreted in a Bayesian sense as the proportion of belief that we assign to each model as being a best fitting model, with the strength of this belief split into a trichotomy by the $E_1$, $E_2$, and $E_3$ events. The

only other model that has non-negligible values for $\mathbb{P}(E_1)$ is the delayed generalised Porous-FKPP model, Eq (6). The remaining models typically have much greater PDE errors than models (3) and (6), especially for models (2) and (4) which do not include a delay. We also see that, while the optimal model does not appear to be dependent on the initial cell density, the errors do seem to decrease with the initial cell density. In S3 Figs I–O in S1 Document and S3 Figs P–U in S1 Document we show, for each data set and for models (3) and (6), respectively, kernel density estimates for each coefficient, plots of the functional forms for each mechanism along with the uncertainty in these functional forms, and solutions to the PDEs with uncertainty. Additionally, our results are consistent with Jin et al. [13] as we also see that $\beta_1$ in (3), where delay is included, is still dependent on the initial cell density. Moreover, the estimates from Jin et al. [13] for $\beta_1$ in their Fisher-Kolmogorov model without delay are all captured within our confidence intervals, although we obtain different estimates for $\gamma_1$ due to this delay mechanism. For the 10,000, 12,000, and 16,000 CPW data sets, our estimates for $\beta_2$ in the delayed Porous-Fisher model (5) also contain the estimates of $\beta_2$ from the Porous-Fisher models fit by Jin et al. [13] where no delay is considered. Finally, as discussed at the start of Section 2, we can now make several comments regarding the difference between the suitability of linear and nonlinear diffusion. Firstly, the model with linear diffusion (3) has been selected more often than the nonlinear diffusion model (6). Second, we note that model (6) can be interpreted as a linear combination of a linear and nonlinear diffusion term. The model simplifies to linear diffusion when $\beta_2 = 0$ or $\beta_3 = 0$. Our results suggest that the confidence intervals for our estimate of $\beta_3$ are greater than unity, but that the confidence interval for $\beta_2$ contains zero in all but two cases. Overall we interpret these results as suggesting that linear diffusion provides the simplest explanation of the data (further discussions are provided in the next paragraph for the hypothesis (10)). Therefore, we see that incorporating uncertainty quantification into our approach is essential for us to make a meaningful distinction between the roles of linear or nonlinear diffusion in these experiments.

We can also make use of the results in Table 1 to perform several hypothesis tests using our uncertainty quantification. For example, for (6) we may be interested in testing the hypotheses

$$H_0 : \beta_2 = 0 \quad \text{versus} \quad H_1 : \beta_2 \neq 0. \tag{10}$$

The null hypothesis $H_0$ in (10) considers the diffusivity to be independent of the cell density, whereas the alternative hypothesis $H_1$ assumes that the diffusivity does depend on this cell density. For all data sets except the 12,000 and 20,000 CPW data sets, the confidence interval for $\beta_2$ in (6) contains zero, suggesting that we would fail to reject the null hypothesis $H_0$ for these data sets at a 95% significance level. This implies that the delayed generalised Porous-FKPP model (6) reduces to the delayed Fisher-Kolmogorov model (3) for this data set. We show this visually in S3 Figs P–U in S1 Document which illustrates that the nonlinear diffusivity curves for model (6) are typically flat and independent of $u$ for smaller densities, only decreasing for larger densities. These larger densities are typically only observed in the data at $t = 48$ h, or only near the edge of the domain $x = 0$ or $x = 1900$ μm, at earlier times. Therefore, these nonlinear diffusivity functions at larger densities may simply be a consequence of having fewer data points at these large density values. Note that the uncertainty in these curves at these larger densities are consistent with having a constant diffusivity $D(u; \boldsymbol{\beta}) = \beta_1$. Only for the 12,000 and 20,000 CPW data sets is there strong evidence to reject $H_0$, since the confidence interval for $\beta_2$ does not contain zero for these two data sets, suggesting a significant difference between these initial cell densities to the other data sets. Moreover, for this 20,000 CPW data set we observe in S3 Fig U in S1 Document. that the nonlinear diffusivity curve for this model is initially decreasing, showing no resemblance to a constant diffusivity $D(u; \boldsymbol{\beta}) = \beta_1$ unlike the

other data sets. We could also test $H_0: \alpha_2 = 0$ for the models with delay, which would imply that no delay is present since the delay mechanism would be constant in time, and we would find for each model that we fail to reject $H_0$, meaning there is evidence that the delay term is needed in these models.

The results for model (6) are worth discussing further as this is the model presented by Lagergren et al. [12]. First, we note that all the $D(u; \boldsymbol{\beta})$ functions are decreasing for this model, in contrast to the nonlinear diffusivity functions found by Lagergren et al. [12], who constrain $D(u; \boldsymbol{\beta})$ to be increasing. Secondly, for most of the data sets our confidence intervals for $\alpha_2$, $\beta_1$, and $\gamma_1$ contain the estimates reported by Lagergren et al. [12], and similarly for $\alpha_1$ for some data sets. The values for $\beta_2$ and $\beta_3$ from Lagergren et al. [12] are not recovered in any of the models as we would expect based on the first point. If we could not have quantified the uncertainty in these models, we might not have observed these similarities or even that we cannot distinguish $\beta_2$ from zero for most of these data sets. An additional result from this uncertainty quantification is that we see that, unlike the models given by Lagergren et al. [12], there is no need to specify that $D(u; \boldsymbol{\beta})$ is monotonic, and should instead be allowed to both increase and decrease over the interval of interest $0 < u < K$; this is not in contradiction to the results of Lagergren et al. [12], as this choice was made in their work out of convenience rather than any particular biological motivation. This more general framework does not preclude the possibility that $D(u; \boldsymbol{\beta})$ is monotonic, but it encompasses a wider range of possibilities.

In Fig 6 we show the PDE solutions for each data set for the delayed Fisher-Kolmogorov model (3), along with the uncertainty around each solution at each time $t$. These PDEs, as discussed in Section 3, are solved using the `DifferentialEquations` package in JULIA [39, 40], and the initial conditions come from sampling the fitted Gaussian process at $t = 0$ h. We

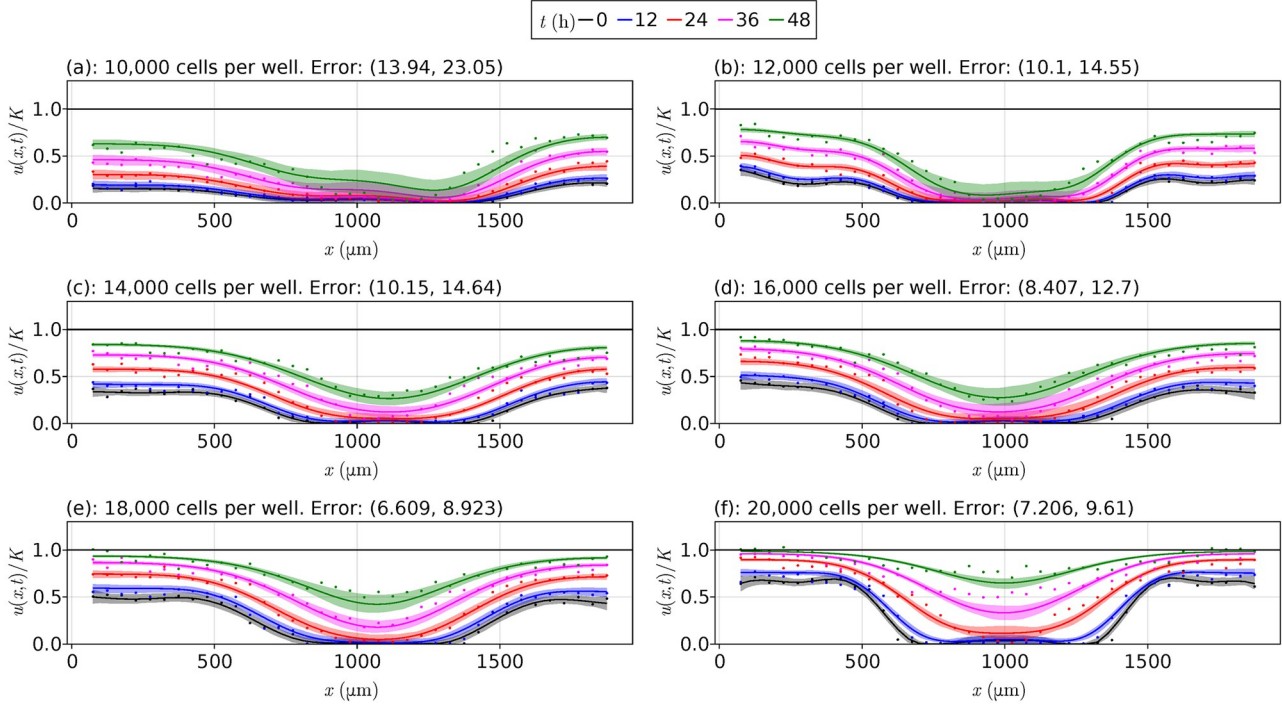

**Fig 6. Optimal results for each data set.** PDE solutions for each data set from Jin et al. [13] for the delayed Fisher-Kolmogorov model (3) from Table 1, with the vertical axis giving $u/K$ rather than $u$, where $K = 1.7 \times 10^{-3}$ cells $\mu m^{-2}$. The points show the data from Jin et al. [13], the solid lines show the mean solution at each corresponding time, and the shaded regions around these lines give the 95% confidence intervals for each solution. Initial conditions are obtained from sampling the Gaussian process rather than using splines. The black horizontal line is at $u/K = 1$.

see that the uncertainty around each solution is small towards the endpoints of the well $x = 0$ μm and $x = 1900$ μm, but increases towards the centre, capturing most of the data points. We again note that these solutions are aimed at learning the underlying function governing the cell density rather than completely recovering the data points, so the uncertainty is not necessarily expected to completely capture all data points. The only model that appears to not be a good fit is for the 20,000 CPW data set in Fig 6(f) as the density curves consistently underestimate the cell densities in the centre of the well. There are at least two clear reasons why these results fail for the 20,000 CPW data set and not the others. Firstly, the cell densities in this data set get much closer to the carrying capacity density than in the other data sets, potentially leading to a depletion of nutrients in the well, implying the need for an additional term in (1) for this nutrient mechanism; in particular, these results demonstrate model misspecification. A second reason could be the noise model, noting that Lagergren et al. [12] fit (6) to this 20,000 CPW data set and find a good fit, in contrast to our results. Lagergren et al. [12] assume the noise is proportional to the data itself at each point and time, whereas we assume the noise is additive, meaning the variance is constant in space and time rather than dependent, and this may only become a problem for this data set due to this depletion of nutrients. We later discuss recommendations for remedying these issues and suggestions for future work.

## 2.4 Conclusions and future work

In this work we present a Gaussian process framework with parametric bootstrapping to perform mechanism discovery with uncertainty quantification for model coefficients, functional forms, and learned dynamical systems governing a biological process. This method is suitable for sparse and noisy data sets and is efficient and parallelisable, which is very different to more standard Bayesian approaches which often rely on inherently serial sampling schemes, such as MCMC [22]. The framework additionally provides a natural way for performing hypothesis testing on the structure of individual mechanisms, and for model comparison and selection with uncertainty. Extending on the work by Jin et al. [13] and Lagergren et al. [12], we demonstrate our methodology on a canonical scratch assay experiment and show that we can learn mechanisms that fit the experimental data well, along with quantified uncertainty for all estimates, for all but one data set, and further support the need for a delay mechanism in interpreting this experiment. We provide evidence that the data reported by Jin et al. [13] is most simply explained by a linear diffusion mechanism rather than a nonlinear diffusion mechanism, and the key to this finding is that our approach involves estimating uncertainty. This is an important finding since the question of determining whether collective cell migration is best described by linear or nonlinear diffusion has remained unresolved and of on-going interest after more than 30 years of investigation [30, 31]. Throughout this analysis we hold the carrying capacity density, $K$, constant, as it is the only parameter of interest that can be directly measured from the data. Future work could involve repeating this work and estimating $K$ along with the other parameters in the mathematical model. This additional parameter would increase the computational requirements to learn the parameters, but the robustness of the results will remain the same. A similar example which supports $K$ being reasonably constant and simple to estimate is given by Browning et al. [41] who, using profile likelihood methods, consider different data sets from a tissue growth experiment and find that $K$ is virtually constant across different experimental conditions, like our experiments that have different initial cell densities.

The data considered in this work are sparse and noisy. One option for future consideration is to explore how de-noising may affect our results, by applying, for example, the neural network methods used by Zhang and Liu [8]. A further way to deal with this noise is to use data

thresholding, where we discard points during a bootstrap iteration that have too low or too high densities, or are evolving too slowly or rapidly. We give an example of data thresholding in S4 Text in S1 Document, and in particular S4 Tables C and D in S1 Document, where we perform a simulation study to demonstrate the effects of data thresholding for models with and without delay. We find that the PDE errors can be decreased with these approaches. These results appear to be data dependent, though, and thresholds are difficult to specify. Future work could extend on these ideas by performing a more robust study of data thresholding, and provide guidelines for identifying these percentages.

In this work we only considered a small number of the most fundamental models. For the nonlinear diffusivity function, $D(u; \boldsymbol{\beta})$, we considered Fisher-Kolmogorov, Porous-Fisher, and generalised Porous-FKPP type models of cell migration. The delay mechanism, $T(t; \boldsymbol{\alpha})$, was assumed to either not be present or take a logistic form. We also assumed a logistic model for the proliferation mechanism $R(u; \boldsymbol{\gamma})$. These are not the only models that could be considered. Motivated by logistic regression, other delay models like the probit and complementary log-log models

$$T(t; \boldsymbol{\alpha}) = \Phi(\alpha_1 + \alpha_2 t) \quad \text{and} \quad T(t; \boldsymbol{\alpha}) = 1 - \exp(-\exp(\alpha_1 + \alpha_2 t)), \qquad (11)$$

respectively, could be used, where $\Phi$ is the cumulative distribution function of the standard normal distribution. We also assume that the delay mechanism is the same for cell migration and proliferation. Future work could extend on this assumption and consider the validity of this assumption for different experiments; the uncertainty quantification from our approach could allow for hypothesis testing to test whether the two mechanisms can be distinguished from each other or are the same for cell migration and proliferation. We could also consider different cell migration models. We observed from the model (6) that the nonlinear diffusivity function appears to be independent of density for sufficiently small densities, and then decreases for larger densities. One model that could effectively model this transition could be a model of the form

$$D(u; \boldsymbol{\beta}) = \beta_1 + \beta_2 \left( \frac{u}{K} \right)^{\beta_3} + \beta_4 \left( \frac{u}{K} \right)^{\beta_5}, \qquad (12)$$

as suggested by Lagergren et al. [12], although this may be difficult to work with because the increase in the number of parameters may lead to potential identifiability issues. For the reaction term, the logistic model appears to work very well for the data set, but we could consider for example the Gompertz and Richards' models, given respectively by [36]

$$R(u; \boldsymbol{\gamma}) = \gamma_1 u \log\left( \frac{K}{u} \right) \quad \text{and} \quad R(u; \boldsymbol{\gamma}) = \gamma_1 u \left[ 1 - \left( \frac{u}{K} \right)^{\gamma_2} \right], \qquad (13)$$

although these two models may also involve issues with parameter identifiability [42]. Future work could consider these types of models and how they fit the data considered in this manuscript. A more valuable study would be to perform a parameter identifiability study for these models to see what types of models can be fit using our methods, providing guidelines for how complex a model can be for our methods to remain robust. We note that our current framework does provide tools for studying the identifiability of parameter estimates through the kernel density estimates of our parameters. In particular, the flatness of the estimated density for a parameter may reveal issues with the parameter's identifiability. This is discussed more in S1 Text in S1 Document. Finally, there may be missing mechanisms in our models. We found that the cells in the 20,000 CPW data set from Jin et al. [13] may approach the carrying capacity density too quickly for the model (1) to be a good fit, potentially implying a missing advection

term or a misspecified noise model at these density values. Future work could expand on this and consider fitting a broader class of models to this data, using our uncertainty quantification to test whether other terms contribute significantly to the model relative to (1).

The optimisation methods used in this manuscript for performing parameter estimation can be improved in several ways. Firstly, in this manuscript we found that the two loss functions (34) and (33) were on two different scales, requiring that we take the logarithm of them separately and then re-combine them. This method for handling the scales may not always work with any experiment. One remedy for this issue would be to consider multi-objective optimisation [43, Chapter 12]. Further extensions to parameter estimation could also be considered, such as an iterative approach with one mechanism being optimised at a time. Finally, future work could consider better methods for obtaining initial parameter estimates without simply running our method for a small number of bootstrap iterations as this approach can be slow. The basis function approach presented in S2 Text in S1 Document provides a remedy for this problem for linear optimisation problems, but extensions to these ideas for nonlinear problems are required.

The Gaussian processes used in this work were kept relatively simple. There are many ways to improve these Gaussian processes and provide better fitting models. More advanced sampling and hyperparameter estimation methods could be used, such as MCMC algorithms. Likelihoods other than Gaussian might be considered for different data types, or we may consider different priors on the hyperparameters [44]. We could also use more advanced kernels, using for example automatic kernel methods [45]. Finally, future work could make the assumed noise model data-dependent, implementing heteroscedastic Gaussian processes [46]. Heteroscedastic Gaussian processes may improve the issues observed in Fig 6 for the 20,000 CPW data set.

Finally, while we only consider models of the form in (1) in this work, in theory any model could be used. By simply including an extra spatial variable in the Gaussian processes, we would be able to apply our ideas to two-dimensional problems like those of Browning et al. [41]. We could also consider more general models such as delay differential equations [47], ordinary differential equation models [48], and stochastic differential equation models [49]. Systems of PDEs could also be considered, as could systems of more general differential equations that may require Gaussian processes with multiple response variables [50]. The choices required to implement these extensions, or to work with other models, are similar to the choices that we describe in this work. For example, the methods we describe for comparing models and optimising the loss function could be directly applied. The main choice that may require some thought is the choice of mean and kernel for the Gaussian process, and similarly the noise model used with the Gaussian process. We use a zero mean function with a squared exponential kernel in this work so that we can model interaction forces for our data, but choices such as the stationary Matèrn kernel or periodic kernel [51] could also be used in future extensions of this work, potentially choosing between kernels using model selection methods [52] or automatic methods [45]. Heteroscedasticity could also be important for other applications [51], which may also change the form of the loss function $\mathcal{L}_{\mathrm{GLS}}$ when incorporating a new scale factor for different times. These considerations highlight another potential advantage over Bayesian methods, which may require substantial thought when changing to a new model or a new application, such as the selection of priors and sampling scheme [26], and the time spent for example configuring a new MCMC scheme could be substantial compared to the time spent choosing a new kernel with our method which will be significantly faster.

## 3 Methods

In this section we describe the methods used for producing the results in this manuscript. We start by describing some pre-processing steps applied to the data. Next, methods for fitting the

Gaussian process and estimating function values and derivatives are discussed. We then discuss the procedure for estimating parameters and using bootstrapping to quantify uncertainty. We then describe how we solve the PDEs numerically. Finally, our methodology for comparing models is discussed. These methods, and all code for the results in this manuscript, are all provided in our JULIA package https://github.com/DanielVandH/EquationLearning.jl.

### 3.1 Data pre-processing

The data sets from Jin et al. [13] have $x = \mathcal{O}(10^3)$ μm, $t = \mathcal{O}(10)$ h, and $u = \mathcal{O}(10^{-3})$ cells μm$^{-2}$. This difference in scales, especially between $x$ and $t$, causes problems with the optimisation routines used in our study. In the implementation of our methods, we rescale $x$ and $t$ such that they are in units of mm and d, respectively, meaning we divide $x$ by $10^3$, $t$ by 24, and multiply $u$ by $10^6$. This rescaling gives $x = \mathcal{O}(1)$ mm, $t = \mathcal{O}(1)$ h, and $u = \mathcal{O}(10^3)$ cells μm$^{-2}$. Additionally, the left-most point in each data set, and for each experimental replicate, is removed as it consistently has a much greater density than the adjacent points, as can be seen from Fig 3.

### 3.2 Fitting Gaussian processes

Suppose we have some spatial grid of $N$ points $x_1, \ldots, x_N$ and a temporal grid of $M$ points $t_1, \ldots, t_M$, where $x_1 < \cdots < x_N$ and $t_1 < \cdots < t_M$. For each $(x_i, t_j)$ on the spatiotemporal grid defined by these points we associate a cell density $u(x_i, t_j) = u_{ij}$. We then define $\mathbf{u} \in \mathbb{R}^{NM \times 1}$ and $\mathbf{X} \in \mathbb{R}^{2 \times NM}$ by

$$
\mathbf{u} = \begin{bmatrix} u_{11} \\ u_{21} \\ \vdots \\ u_{N1} \\ u_{12} \\ \vdots \\ u_{NM} \end{bmatrix} \quad \text{and} \quad \mathbf{X} = \begin{bmatrix} x_1 & x_2 & \cdots & x_N & x_1 & x_2 & \cdots & x_N & \cdots & x_N \\ t_1 & t_1 & \cdots & t_1 & t_2 & t_2 & \cdots & t_2 & \cdots & t_M \end{bmatrix}. \tag{14}
$$

We note that (14) is written for only a single experimental replicate for simplicity; for more experimental replicates, as we have in the data from Jin et al. [13], the only change is that we simply extend $\mathbf{u}$ and $\mathbf{X}$ with these additional data points and corresponding grid points, noting that the $(x_i, t_j)$ pairs do not have to be unique. Following Rasmussen and Williams [33], let us assume that there is some function $\mathbf{u}^*$ that describes our cell densities. We model the experimentally observed densities $\mathbf{u}$ according to $\mathbf{u} = \mathbf{u}^* + \sigma_n \mathbf{z}$, where $\mathbf{z} \sim \mathcal{N}(\mathbf{0}, \mathbf{I})$, where $\mathbf{I}$ is the identity matrix and $\sigma_n$ is the standard deviation of the observation noise. We let $\mathbf{u}^*$ be given by the Gaussian process

$$
\mathbf{u}^* \sim \mathcal{GP}\left(\mathbf{0}, \mathbf{K}(\mathbf{X}, \mathbf{X})\right). \tag{15}
$$

Here, the mean function is taken to be zero as this only shifts the conditional mean [53] which the kernel $\mathbf{K}(\mathbf{X}, \mathbf{X})$ can easily adjust for [54]; note that the mean function being zero does not imply that we assume the densities have zero mean as the conditional mean of the data will be non-zero, as shown later in (18). The covariance function $\mathbf{K}(\mathbf{P}, \mathbf{Q})$ denotes the kernel matrix between $\mathbf{P}$ and $\mathbf{Q}$, with the $(i, j)$th entry giving the covariance between the $i$th column of $\mathbf{P}$, $\mathbf{p}_i$, and the $j$th column of $\mathbf{Q}$, $\mathbf{q}_j$, written as $k(\mathbf{p}_i, \mathbf{q}_j)$. There are many possible choices for the kernel

$k$ [33, Chapter 4] and in this manuscript we only consider the squared-exponential kernel,

$$k(\mathbf{p}_i, \mathbf{q}_j) = \sigma_f^2 \exp\left\{ -\frac{1}{2}\left(\mathbf{p}_i - \mathbf{q}_j\right)^\top \mathbf{\Lambda}^{-1}\left(\mathbf{p}_i - \mathbf{q}_j\right) \right\}, \tag{16}$$

where $\sigma_f$ is the standard deviation of the noise-free data and $\mathbf{\Lambda} = \mathrm{diag}(\ell_1^2, \ell_2^2)$, where $\ell_1$ and $\ell_2$ are the spatial and temporal scales, respectively. This kernel (16) is chosen so that we model the interactive forces between cells as being stronger when they are close, and negligible when they are far enough away from each other.

The hyperparameters $\sigma_n$, $\sigma_f$, $\ell_1$, and $\ell_2$ are estimated in JULIA [40] using the `Gaussian-Processes` package developed by Fairbrother et al. [55]. This package estimates the hyperparameters using maximum likelihood estimation [33, Chapter 5] using the `Optim` package by Mogensen et al. [56]. The L-BFGS algorithm is used in `Optim`. This algorithm requires initial estimates for each parameter. We choose rough bounds for each parameter to sample initial estimates between, although we keep the optimisation unconstrained. We then use the `LatinHypercubeSampling` package [57] to generate $w$ initial values for each hyperparameter. For the resulting $w$ sets of hyperparameters from each optimisation, we choose the set which gives the largest marginal likelihood. We use $w = 250$ in this work and we sample the hyperparameters according to $\ell_1 \in [10^{-6}, 1]$, $\ell_2 \in [10^{-6}, 1]$, $\sigma_f \in [10^{-6}, 7\,\mathbb{SD}(u)]$, and $\sigma_n \in [10^{-6}, 7\,\mathbb{SD}(u)]$, where $\mathbb{SD}(u)$ denotes an estimate of the standard deviation of the density data $u$; these bounds define the hypercube used with the `LatinHypercubeSampling` package. These length scale bounds are used as we estimate the data between 0 and 1, as discussed later. We note that the final optimised hyperparameters are not enforced to stay within these bounds. This fitting of the Gaussian process can be the most time consuming part of the method, depending on how many restarts $w$ are needed for obtaining reliable parameter values; this value $w$ could be small, though, for data sets with less noise or for more carefully chosen initial values for $\ell_1$, $\ell_2$, $\sigma_f$, and $\sigma_n$.

With our observed data $\mathbf{u}$ at the spatiotemporal points $\mathbf{X}$, let us assume that we are interested in evaluating the Gaussian process at some other points $x_1^*, \ldots, x_n^*$ and $t_1^*, \ldots, t_m^*$ in space and time respectively, and that we have obtained optimal values for the hyperparameters. We use these points to define a matrix $\mathbf{X}^* \in \mathbb{R}^{2 \times nm}$, where the rows are defined similarly to those in $\mathbf{X}$ in (14). The samples from the Gaussian process, $\mathbf{u}^*$, are then given by a $\mathbb{R}^{nm \times 1}$ vector, and we write [33]

$$\mathbf{u}^* \mid \mathbf{X}, \mathbf{u}, \mathbf{X}^* \sim \mathcal{N}(\bar{\mathbf{u}}^*, \mathbb{C}\mathrm{ov}(\mathbf{u}^*)), \tag{17}$$

where the mean and covariance are given by

$$\bar{\mathbf{u}}^* = \mathbf{K}(\mathbf{X}^*, \mathbf{X})[\mathbf{K}(\mathbf{X}, \mathbf{X}) + \sigma_n^2 \mathbf{I}_{NM}]^{-1}\mathbf{u}, \tag{18}$$

$$\mathbb{C}\mathrm{ov}(\mathbf{u}^*) = \mathbf{K}(\mathbf{X}^*, \mathbf{X}^*) - \mathbf{K}(\mathbf{X}^*, \mathbf{X})[\mathbf{K}(\mathbf{X}, \mathbf{X}) + \sigma_n^2 \mathbf{I}_{NM}]^{-1}\mathbf{K}(\mathbf{X}, \mathbf{X}^*), \tag{19}$$

where $\mathbf{I}_{NM} \in \mathbb{R}^{NM \times NM}$ is the $NM$-square identity matrix. We remark that the covariance omits an extra noise term that would be used to estimate the noisy data [33, p. 19] since we are interested only in the underlying function describing the cell density profile as a function of space and time rather than the actual values of the values of the data when considering noise, i.e. we are modelling $\mathbf{u}^*$ rather than $\mathbf{u}$.

In this work we choose the $x_i^*$ to be $n = 50$ equally spaced points between $x_1$ and $x_N$, and the $t_j^*$ are 50 equally spaced points between $t_1$ and $t_M$, meaning $x_i^* = x_1 + (i - 1)\Delta x^*$ for $i = 1, \ldots,$ $n$ and $t_j^* = t_j + (j - 1)\Delta t^*$ for $j = 1, \ldots, m$, where $\Delta x^* = (x_N - x_1)/(n - 1)$ and $\Delta t^* = (t_M - t_1)/$

$(m-1)$. With these choices of $x_i^*$ and $t_j^*$, the optimisation problem was unstable due to the difference in spatial and temporal scales. To stabilise the estimates, we normalise the data to be between 0 and 1, defining

$$\xi_i = \frac{x_i^* - x_1}{x_N - x_1} \quad \text{and} \quad \tau_j = \frac{t_j^* - t_1}{t_M - t_1}, \quad i = 1, \ldots, n, \ j = 1, \ldots, m. \tag{20}$$

We replace the corresponding entries in $\mathbf{X}^*$ with these scaled entries, and we apply the same scaling to the data $(x_i, t_j)$ that then replaces the corresponding entries in $\mathbf{X}$.

### 3.3 Estimating function values and derivatives

Following the fitting of the Gaussian process, we need estimates for the cell density and the corresponding derivative. We can find these derivatives by noting that for a Gaussian process

$$f(\mathbf{x}) \sim \mathcal{GP}(\mathbf{0}, k(\mathbf{x}, \mathbf{x}')), \tag{21}$$

we have

$$Lf(\mathbf{x}) \sim \mathcal{GP}(0, L_1 L_2 k(\mathbf{x}, \mathbf{x}')), \tag{22}$$

where $L$ is a linear differential operator and $L_i$ denotes $L$ applied to the $i$th argument of $k(\mathbf{x}, \mathbf{x}')$ [14]. Using similar ideas to those used by Wang et al. [14], we can show that

$$\begin{bmatrix} \mathbf{u}^* \\ \dfrac{\partial \mathbf{u}^*}{\partial \tau} \\ \dfrac{\partial \mathbf{u}^*}{\partial \xi} \\ \dfrac{\partial^2 \mathbf{u}^*}{\partial \xi^2} \end{bmatrix} \mid \mathbf{X}, \mathbf{u}, \mathbf{X}^* \sim \mathcal{N}(\boldsymbol{\mu}, \boldsymbol{\Sigma}), \tag{23}$$

where

$$\mathbf{M} = \begin{bmatrix} \mathbb{C}\mathrm{ov}(\mathbf{u}^*, \mathbf{u}) \\ \mathbb{C}\mathrm{ov}\left(\dfrac{\partial \mathbf{u}^*}{\partial \tau}, \mathbf{u}\right) \\ \mathbb{C}\mathrm{ov}\left(\dfrac{\partial \mathbf{u}^*}{\partial \xi}, \mathbf{u}\right) \\ \mathbb{C}\mathrm{ov}\left(\dfrac{\partial^2 \mathbf{u}^*}{\partial \xi^2}, \mathbf{u}\right) \end{bmatrix}, \tag{24}$$

$$\mathbf{K}^* = [\mathbf{K}(\mathbf{X}, \mathbf{X}) + \sigma_n^2 \mathbf{I}_{NM}]^{-1}, \tag{25}$$

$$\boldsymbol{\mu} = \mathbf{M}\mathbf{K}^* \mathbf{u}, \tag{26}$$

$$\boldsymbol{\Sigma} = \begin{bmatrix} \mathbb{C}\text{ov}(\mathbf{u}^*, \mathbf{u}^*) & \mathbb{C}\text{ov}\left(\mathbf{u}^*, \frac{\partial \mathbf{u}^*}{\partial \tau}\right) & \mathbb{C}\text{ov}\left(\mathbf{u}^*, \frac{\partial \mathbf{u}^*}{\partial \xi}\right) & \mathbb{C}\text{ov}\left(\mathbf{u}^*, \frac{\partial^2 \mathbf{u}^*}{\partial \xi^2}\right) \\ \mathbb{C}\text{ov}\left(\frac{\partial \mathbf{u}^*}{\partial \tau}, \mathbf{u}^*\right) & \mathbb{C}\text{ov}\left(\frac{\partial \mathbf{u}^*}{\partial \tau}, \frac{\partial \mathbf{u}^*}{\partial \tau}\right) & \mathbb{C}\text{ov}\left(\frac{\partial \mathbf{u}^*}{\partial \tau}, \frac{\partial \mathbf{u}^*}{\partial \xi}\right) & \mathbb{C}\text{ov}\left(\frac{\partial \mathbf{u}^*}{\partial \tau}, \frac{\partial^2 \mathbf{u}^*}{\partial \xi^2}\right) \\ \mathbb{C}\text{ov}\left(\frac{\partial \mathbf{u}^*}{\partial \xi}, \mathbf{u}^*\right) & \mathbb{C}\text{ov}\left(\frac{\partial \mathbf{u}^*}{\partial \xi}, \frac{\partial \mathbf{u}^*}{\partial \tau}\right) & \mathbb{C}\text{ov}\left(\frac{\partial \mathbf{u}^*}{\partial \xi}, \frac{\partial \mathbf{u}^*}{\partial \xi}\right) & \mathbb{C}\text{ov}\left(\frac{\partial \mathbf{u}^*}{\partial \xi}, \frac{\partial^2 \mathbf{u}^*}{\partial \xi^2}\right) \\ \mathbb{C}\text{ov}\left(\frac{\partial^2 \mathbf{u}^*}{\partial \xi^2}, \mathbf{u}^*\right) & \mathbb{C}\text{ov}\left(\frac{\partial^2 \mathbf{u}^*}{\partial \xi^2}, \frac{\partial \mathbf{u}^*}{\partial \tau}\right) & \mathbb{C}\text{ov}\left(\frac{\partial^2 \mathbf{u}^*}{\partial \xi^2}, \frac{\partial \mathbf{u}^*}{\partial \xi}\right) & \mathbb{C}\text{ov}\left(\frac{\partial^2 \mathbf{u}^*}{\partial \xi^2}, \frac{\partial^2 \mathbf{u}^*}{\partial \xi^2}\right) \end{bmatrix}^{-1} \tag{27}$$
$$- \mathbf{M}\mathbf{K}^*\mathbf{M}^\top.$$

Note that these derivatives are on the scale of the data that we scaled between 0 and 1 in (20). These derivatives can be put onto the correct scale by dividing by the denominators in (20), for example the temporal derivatives $\partial/\partial\tau$ are related to $\partial/\partial t$ by $\partial/\partial t = [1/(t_M - t_1)]\partial/\partial\tau$ and similarly $\partial^2/\partial x^2 = [1/(x_N - x_1)^2]\partial^2/\partial\xi^2$. The covariances in (24)–(27) are computed using the property in (22), for example

$$\mathbb{C}\text{ov}\left(\mathbf{u}, \frac{\partial \mathbf{u}^*}{\partial \tau}\right) = \left\{\frac{\partial}{\partial \tau} k\left(\mathbf{x}_i, \mathbf{x}_j^*\right)\right\}_{\substack{i=1,\dots,NM, \\ j=1,\dots,nm}} \tag{28}$$

and

$$\mathbb{C}\text{ov}\left(\frac{\partial^2 \mathbf{u}^*}{\partial \xi^2}, \frac{\partial \mathbf{u}^*}{\partial \tau}\right) = \left\{\frac{\partial^3}{\partial \xi_i^2 \partial \tau_j} k\left(\mathbf{x}_i^*, \mathbf{x}_j^*\right)\right\}_{i,j=1,\dots,nm}, \tag{29}$$

where $NM$ and $nm$ are the number of experimental replicates and test points, respectively, with $k$ as defined in (16), and we denote by $\mathbf{x}_i$ and $\mathbf{x}_j^*$ the $i$th and $j$th columns of $\mathbf{X}$ and $\mathbf{X}^*$, respectively.

To now use (23) to sample the function values and corresponding derivatives, let $\mathbf{U}$ be the random vector on the left of (23). We then write $\mathbf{U} = \boldsymbol{\mu} + \mathbf{L}\mathbf{z}$, where $\mathbf{L}$ is the Cholesky factor of $\boldsymbol{\Sigma}$ such that $\mathbf{L}\mathbf{L}^\top = \boldsymbol{\Sigma}$, and $\mathbf{z} \sim \mathcal{N}(\mathbf{0}, \mathbf{I})$. Drawing a sample $\mathbf{z}$ from $\mathcal{N}(\mathbf{0}, \mathbf{I})$ thus constitutes a draw from $\mathbf{U}$ from which we can obtain the function values and derivatives. All Cholesky decompositions are computed using the `cholesky` function from the `LinearAlgebra` package in JULIA. A common issue when building covariance matrices, due to numerical issues related to the matrix's eigenvalues, is that the matrix $\boldsymbol{\Sigma}$ may not be symmetric positive definite [33]. Hence, for these Cholesky factors we need to make a numerical adjustment to ensure that $\boldsymbol{\Sigma}$ is symmetric positive definite. To do so, we follow the approach by Chen et al. [15]. In particular, let us write the diagonal entries of $\boldsymbol{\Sigma}$ as

$$\begin{aligned} \text{diag}(\boldsymbol{\Sigma}) &= \text{diag}\left[\mathbb{C}\text{ov}(\mathbf{u}^*, \mathbf{u}^*), \mathbb{C}\text{ov}\left(\frac{\partial \mathbf{u}^*}{\partial \tau}, \frac{\partial \mathbf{u}^*}{\partial \tau}\right), \right. \\ &\quad \left. \mathbb{C}\text{ov}\left(\frac{\partial \mathbf{u}^*}{\partial \xi}, \frac{\partial \mathbf{u}^*}{\partial \xi}\right), \mathbb{C}\text{ov}\left(\frac{\partial^2 \mathbf{u}^*}{\partial \xi^2}, \frac{\partial^2 \mathbf{u}^*}{\partial \xi^2}\right)\right] \\ &= \text{diag}(\boldsymbol{\Sigma}_{11}, \boldsymbol{\Sigma}_{22}, \boldsymbol{\Sigma}_{33}, \boldsymbol{\Sigma}_{44}). \end{aligned} \tag{30}$$

Letting tr($\mathbf{A}$) denote the trace of a matrix $\mathbf{A}$, we define the nugget terms

$$\eta_1 = \eta, \quad \eta_2 = \frac{\text{tr}(\mathbf{\Sigma}_{22})}{\text{tr}(\mathbf{\Sigma}_{11})}\eta, \quad \eta_3 = \frac{\text{tr}(\mathbf{\Sigma}_{33})}{\text{tr}(\mathbf{\Sigma}_{11})}\eta, \quad \text{and} \quad \eta_4 = \frac{\text{tr}(\mathbf{\Sigma}_{44})}{\text{tr}(\mathbf{\Sigma}_{11})}\eta, \tag{31}$$

where $\eta = 10^{-5}$ in this manuscript. We then define $\boldsymbol{\eta} = \text{diag}(\eta_1\mathbf{I}_{nm}, \eta_2\mathbf{I}_{nm}, \eta_3\mathbf{I}_{nm}, \eta_4\mathbf{I}_{nm})$ and replace $\Sigma$ with $\Sigma + \boldsymbol{\eta}$. This procedure for regularising $\Sigma$ to ensure it remains symmetric positive definite allows the nugget terms to account for the order of the corresponding derivative term relative to the function values.

## 3.4 Parameter estimation

In this section we assume that we have already drawn some function values $\mathbf{u}^*$ and derivatives $\partial \mathbf{u}^*/\partial t$, $\partial \mathbf{u}^*/\partial x$, and $\partial^2 \mathbf{u}^*/\partial x^2$, where the derivatives have been transformed onto the original scale rather than the unit interval. These values are obtained over the gridpoints $(\xi_i, \tau_j)$ for $i = 1, \ldots, n$ and $j = 1, \ldots, m$. We use these values to estimate each term in (1), where we expand the flux term for $i = 1, \ldots, n$ and $j = 1, \ldots, m$,

$$\frac{\partial u_{ij}^*}{\partial t} = T\left(\tau_j^*; \boldsymbol{\alpha}\right)\left[\frac{\text{d}D\left(u_{ij}^*; \boldsymbol{\beta}\right)}{\text{d}u}\left(\frac{\partial u_{ij}^*}{\partial x}\right)^2 + D\left(u_{ij}^*; \boldsymbol{\beta}\right)\frac{\partial^2 u_{ij}^*}{\partial x^2} + R\left(u_{ij}^*; \boldsymbol{\gamma}\right)\right], \tag{32}$$

where $u_{ij}^* = u^*(\xi_i, \tau_j)$ is the estimate for the cell density from the Gaussian process at $(\xi_i, \tau_j)$, or at $(x_i, t_j)$ after rescaling. Note that (32) requires that $D$ is differentiable at all the gridpoints.

To estimate the parameters $\boldsymbol{\theta} = (\boldsymbol{\alpha}^\top, \boldsymbol{\beta}^\top, \boldsymbol{\gamma}^\top)^\top$ we first define a loss function to be optimised. Following Lagergren et al. [12], we define two loss functions based on (32) and differences between a numerical solution of (1) and the actual data. We define $\hat{u}_{ij}$ to be an estimate of the cell density $u$ at $(x_i, t_j)$ for $i = 1, \ldots, N$ and $j = 1, \ldots, M$. We then define the two loss functions:

$$\mathcal{L}_{\text{PDE}}(\boldsymbol{\theta}) = \frac{1}{nm}\sum_{i=1}^{n}\sum_{j=1}^{m}\left\{\frac{\partial u_{ij}^*}{\partial t} - T\left(t_j^*; \boldsymbol{\alpha}\right)\left[\frac{\text{d}D(u_{ij}^*; \boldsymbol{\beta})}{\text{d}u}\left(\frac{\partial u_{ij}^*}{\partial x}\right)^2\right.\right.$$
$$\left.\left. + D(u_{ij}^*; \boldsymbol{\beta})\frac{\partial^2 u_{ij}^*}{\partial x^2} + R(u_{ij}^*; \boldsymbol{\gamma})\right]\right\}^2, \tag{33}$$

$$\mathcal{L}_{\text{GLS}}(\boldsymbol{\theta}) = \frac{1}{NM}\sum_{i=1}^{N}\sum_{j=1}^{M}w_j\left(\frac{\hat{u}_{ij} - u_{ij}}{\sigma_n}\right)^2. \tag{34}$$

The loss function $\mathcal{L}_{\text{PDE}}(\boldsymbol{\theta})$ helps to ensure that our learned parameters lead to results that correspond to the solution of (1). Furthermore, this loss function contributes most to how we obtain the quantified uncertainty, as this term is where the parameters $\boldsymbol{\theta}$ appear directly. The loss function $\mathcal{L}_{\text{GLS}}(\boldsymbol{\theta})$ ensures that the learned solutions do not go too far from the experimental data. In particular, $\mathcal{L}_{\text{PDE}}$ is used to control the curves relative to the PDE and constrains the sampled values, while $\mathcal{L}_{\text{GLS}}$ is used to control the curves relative to the experimental data. The weights $w_j$ in (34) are $w_1 = 10$ and $w_j = 1$ for $j > 1$ in this manuscript. We include these weights so that the initial condition has an extra weighting compared to the $t > 0$ data, noting that the initial data does not have to satisfy the same governing model (1)—this is the same weighting used by Lagergren et al. [12]; the choice of the hyperparameter $w_1 = 10$ depends on the data, and $w_1 = 1$ is a reasonable default. The values $\hat{u}_{ij}$ are obtained by solving (1) at the given parameter values $\boldsymbol{\theta}$ and taking the solution at the point that is closest to $(x_i, t_j)$, where $i = 1, \ldots, N$ and $j = 1, \ldots, M$. To solve the differential equations we use the JULIA package

DifferentialEquations [40] as described in Section 3.6. The division by $\sigma_n$ in (34) ensures that the residual $\hat{u}_{ij} - u_{ij}$ has constant variance, assuming our Gaussian process holds, motivated by the generalised least squares (GLS) residual developed by Lagergren et al. [12, 58]. To counteract the fact that $\mathcal{L}_{\mathrm{PDE}}$ and $\mathcal{L}_{\mathrm{GLS}}$ are on quite different scales, we take the logarithm of each and define the loss function $\mathcal{L}(\boldsymbol{\theta}) = \log[\mathcal{L}_{\mathrm{PDE}}(\boldsymbol{\theta})] + \log[\mathcal{L}_{\mathrm{GLS}}(\boldsymbol{\theta})]$, allowing the optimiser to more easily converge with these smaller function values. We then obtain estimates $\hat{\boldsymbol{\theta}} = (\hat{\boldsymbol{\alpha}}^\top, \hat{\boldsymbol{\beta}}^\top, \hat{\boldsymbol{\gamma}}^\top)^\top$ by solving the optimisation problem

$$\hat{\boldsymbol{\theta}} = \underset{\boldsymbol{\theta} \in \mathbb{R}^{a+d+r}}{\arg\min} \mathcal{L}(\boldsymbol{\theta}), \tag{35}$$

assuming that $\boldsymbol{\alpha} \in \mathbb{R}^a, \boldsymbol{\beta} \in \mathbb{R}^d$, and $\boldsymbol{\gamma} \in \mathbb{R}^r$. To ensure that the found parameter values lead to physically realistic functional forms, we set the loss to be infinite if there are any negative delay or nonlinear diffusivity function values, or if the integral of the reaction function over $[0, u_M]$, where $u_M$ is the maximum density value, is negative. This integral $\int_0^{u_M} R(u)\mathrm{d}u$ is computed numerically using Gauss-Legendre quadrature, computing the Gauss-Legendre nodes using the FastGaussQuadrature package [59]. We use the Optim package in JULIA [56] to solve this optimisation problem with the L-BFGS algorithm. The Optim package leverages the automatic differentiation package ForwardDiff [60] for computing gradients and Hessians of $\mathcal{L}$ with respect to $\boldsymbol{\theta}$.

The optimisation problem (35) can be very sensitive to the initial values used for each parameter, especially for $\boldsymbol{\beta}$ which is not surprising given the results observed by Jin et al. [13] and convergence problems identified by Kaltenbacher et al. [61] in their fixed point approach to recovering nonlinear terms in reaction-diffusion equations. To overcome this problem we choose a range of initial values within some hypercube based on individual bounds for each parameter. Similar to our optimisation procedure for fitting the Gaussian processes, the JULIA package LatinHypercubeSampling [57] is used for this purpose. In this work we use multiple initial values each with around 10 bootstrap iterations (with bootstrapping defined in the next section) and around 5 initial sets of parameters chosen. Slight variations from 10 and 5 are only used if the results are unclear. Using these 10 results we obtain rough estimates for each parameter $\theta_i$ in $\boldsymbol{\theta}$, $i = 1, \ldots, a + d + r$, which we then use to scale the parameters such that $\theta_i = \mathcal{O}(1)$ for each $i = 1, \ldots, a + d + r$. This scaling puts all parameters on the same scale so that multiple initial values are not typically required, and we may simply start each parameter at $\theta_i = 1$ for $i = 1, \ldots, a + d + r$. This scaling also helps the convergence of the algorithm in general since relative scaling between parameters is a crucial factor in the optimiser's performance [62, Section 7.5].

## 3.5 Bootstrapping and uncertainty quantification

Now we discuss how we implement parametric bootstrapping, providing us with uncertainty quantification for our learned coefficients, functional forms, and solutions to PDEs. We start with the Gaussian process (23) and assume that $\boldsymbol{\mu}$ and $\mathbf{L}$, the Cholesky factor of $\Sigma$, have already been computed. We then draw $B$ random variates $\mathbf{z}^{(b)}$, $b = 1, \ldots, B$, where each $\mathbf{z}^{(b)} \sim \mathcal{N}(\mathbf{0}, \mathbf{I})$. Note that $\boldsymbol{\mu}$ and $\mathbf{L}$ only have to be computed once. We can then compute $\mathbf{U}^{(b)} = \boldsymbol{\mu} + \mathbf{L}\mathbf{z}^{(b)}$ for each $b = 1, \ldots, B$, giving $B$ sets of parameter vectors $\hat{\boldsymbol{\theta}}^{(b)}$ that are solutions to the optimisation problem (35), noting that each $\mathbf{U}^{(b)}$ leads to a different set of function values and derivatives. These results in turn lead to $B$ sets of parameter vectors $\hat{\boldsymbol{\alpha}}^{(b)}, \hat{\boldsymbol{\beta}}^{(b)}$, and $\hat{\boldsymbol{\gamma}}^{(b)}$, for $b = 1, \ldots, B$. In this work we mostly use $B = 100$. This process could easily be parallelised as the parameter estimation for each $b$ is independent of the other bootstrap iterations.

The $B$ samples for each individual parameter $\theta_i$, $i = 1, \ldots, a + d + r$, defines a sampling distribution for that parameter, which we can then use to extract quantiles and density estimates. The `KernelDensity` package in JULIA [63] is used for obtaining kernel density estimates for each parameter. We obtain 95% confidence intervals using JULIA's `quantile` function at levels 2.5% and 97.5%, giving us a sense for the uncertainty in each parameter.

In addition to being able to obtain confidence intervals for parameter estimates, we can obtain confidence intervals for the functional forms themselves by simply computing the corresponding $B$ curves for each learned functional form over some grid of values, and then taking the quantiles at each point. For example, if we had $D(u; \boldsymbol{\beta})$ we could evaluate this function for each $\boldsymbol{\beta}^{(b)}$, $b = 1, \ldots, B$, over some range of values $[u_m, u_M]$ of $u$, say $[0, K]$. These evaluations provide $B$ samples for the nonlinear diffusivity curve on this range. We then take each point used in $[u_m, u_M]$ and compute the 2.5% and 97.5% quantiles, along with the mean of the samples at this point. We can then plot these values to give the mean value of the function at each $u$, and a ribbon plot can be used to demonstrate the uncertainty around these function values. Similar ideas can be used for obtaining confidence intervals for the PDE solutions.

## 3.6 Numerical solution of PDEs

Here we describe the procedure we use for numerically solving (1). The boundary conditions are given by

$$\frac{\partial u(a, t)}{\partial x} = \frac{\partial u(b, t)}{\partial x} = 0, \tag{36}$$

following Jin et al. [13]. These boundary conditions are appropriate as they imply that the distribution of cells is uniform in space, as is clear in Fig 1(a), so that the net movement of cells across the boundaries is zero [64]. We use $a = 75$ μm and $b = 1875$ μm, remembering that we removed the leftmost point at $x = 25$ μm.

The discretisation of the PDE proceeds by using a vertex-centred finite volume method [65]. We place $N_p$ nodes $x_1, \ldots, x_{N_p}$ over some mesh (with these $x_i$ being distinct from the data discussed), with $a = x_1 < x_2 < \cdots < x_{N_p-1} < x_{N_p} = b$ and $a = 75$, $b = 1875$. We assume that these points are equally spaced so that $\Delta x = x_{i+1} - x_i$ for $i = 1, \ldots, N_p - 1$ is constant. In particular, we set $x_j = a + (j-1)\Delta x$ with $\Delta x = (b-a)/(N_p - 1)$ for $j = 1, \ldots, N_p$. We then set

$$
\begin{aligned}
w_i &= \begin{cases} x_1 & i = 1, \\ \frac{1}{2}(x_{i-1} + x_i) & i = 2, \ldots, N_p, \end{cases} \\[2ex]
e_i &= \begin{cases} \frac{1}{2}(x_i + x_{i+1}) & i = 1, \ldots, N_p - 1, \\ x_{N_p} & i = N_p, \end{cases} \\[2ex]
V_i &= \begin{cases} \frac{1}{2}\Delta x & i \in \{1, N_p\}, \\ \Delta x & i = 2, \ldots, N_p - 1. \end{cases}
\end{aligned}
\tag{37}
$$

Next, define the control volume averages

$$\bar{u}_i(t) = \frac{1}{V_i}\int_{w_i}^{e_i} u(x, t)\,\mathrm{d}x \quad \text{and} \quad \bar{R}_i(t) = \frac{1}{V_i}\int_{w_i}^{e_i} R(u(x, t))\,\mathrm{d}x, \quad i = 1, \ldots, N_p, \; t > 0. \tag{38}$$

With these definitions, (1) becomes, for $i = 1, \ldots, N_p$,

$$\frac{d\bar{u}_i}{dt} = T(t)\left\{\frac{1}{V_i}\left[D(u(e_i, t))\frac{\partial u(e_i, t)}{\partial x} - D(u(w_i, t))\frac{\partial u(w_i, t)}{\partial x}\right] + \bar{R}_i\right\}, \quad t > 0. \tag{39}$$

We then make the following approximations and let $u_i = u(x_i, t)$, $R_i = R(u_i)$, and $D_i = D(u_i)$:

$$\bar{u}_i(t) \approx u_i, \quad \bar{R}_i(t) \approx R_i, \quad i = 2, \ldots, N_p, \tag{40}$$

$$D(u(w_i, t)) \approx \frac{D_{i-1} + D_i}{2}, \quad i = 2, \ldots, N_p, \tag{41}$$

$$D(u(e_i, t)) \approx \frac{D_i + D_{i+1}}{2}, \quad i = 1, \ldots, N_p - 1, \tag{42}$$

$$\frac{\partial u(w_i, t)}{\partial x} \approx \frac{u_i - u_{i-1}}{\Delta x}, \quad i = 2, \ldots, N_p \tag{43}$$

$$\frac{\partial u(e_i, t)}{\partial x} \approx \frac{u_{i+1} - u_i}{\Delta x}, \quad i = 1, \ldots, N_p - 1. \tag{44}$$

We thus obtain the following system of ordinary differential equations that approximates (1):

$$
\begin{aligned}
\frac{du_1}{dt} &= T(t)\left\{\frac{2}{\Delta x}\left[\left(\frac{D_1 + D_2}{2}\right)\left(\frac{u_2 - u_1}{\Delta x}\right)\right] + R_1\right\}, \\
\frac{du_i}{dt} &= T(t)\left\{\frac{1}{2\Delta x^2}\left[(D_i + D_{i+1})(u_{i+1} - u_i) - (D_{i-1} + D_i)(u_i - u_{i-1})\right] + R_i\right\}, \\
\frac{du_{N_p}}{dt} &= T(t)\left\{\frac{2}{\Delta x}\left[-\left(\frac{D_{N_p-1} + D_{N_p}}{2}\right)\left(\frac{u_{N_p} - u_{N_p-1}}{\Delta x}\right)\right] + R_{N_p}\right\},
\end{aligned}
\tag{45}
$$

where $i = 2, \ldots, N_p - 1$ in $du_i/dt$. We solve this system of ODEs in (45) using the JULIA package `DifferentialEquations` [40] which computes Jacobians using the `ForwardDiff` package [60]. We take $N_p = 500$ in this work, although the results are not sensitive to this choice as we show in S5 Text in S1 Document, where we argue that the PDE truncation errors are not too important for the bootstrapping procedure in this application. In particular, we obtain essentially the same results in Table 1 with $N_p = 500$ as we do with $N_p = 50$, and the best models selected remain the same, as demonstrated in S5 Tables E–H in S1 Document and S5 Fig V in S1 Document. Hence, we do not need to be concerned about any issues caused by solving these PDEs across a wide range of parameter sets with a fixed number of gridpoints. In addition, on a Windows 10 64-bit machine with an i7–8700K CPU @ 3.70 GHz, 32 GB of RAM, and a GTX 1070 Ti (without parallel computing) the computing time with $N_p = 500$ to produce Table 1 is around 30 hours, while with $N_p = 50$ the computing time is just under one hour.

We now discuss the choice of initial condition $u(x, 0) = f(x)$ for solving (45). One choice is to follow Jin et al. [13] and simply fit a linear spline through the averaged density data at time $t = 0$. This fitting can be accomplished using the implementation of the Dierckx Fortran library [66] in JULIA available from the `Dierckx` package [67]. We refer to this initial condition (IC) as a *Spline IC*.

One other choice is to allow the initial condition to be sampled rather than fixed based on the experimental data, noting that we expect there to be significant noise in the data

considered in this manuscript. Simpson et al. [42] demonstrated in the context of population growth that estimating an initial condition rather than fixing it based on observed data leads to significantly improved results, and we observe similar improvements for this data set by Jin et al. [13], with better fitting curves and variability that captures the data. We refer to this initial condition as a *Sampled IC*.

The initial conditions can be sampled as follows. Suppose we are solving (1) with parameters $\hat{\boldsymbol{\theta}}^{(b)}$ from the $b$th bootstrap iteration. These parameters will correspond to some densities $\mathbf{u}^{*(b)}$. We can then consider the initial condition to be those densities corresponding to $t = 0$. For these densities, we note that the Gaussian processes may give negative values, and we therefore need to replace any negative densities with zero. Extensions to this work may implement constrained Gaussian processes to prohibit any negative values [68]. To allow for a larger mesh for the PDE grid than the $n \times m$ grid used for bootstrapping, we can then use a linear spline to fit through these densities and then evaluate the spline over the PDE mesh. This evaluation defines an initial condition corresponding to the $b$th bootstrap iteration which can then be used to solve the PDE.

Regardless of the choice of initial condition, we end up with $B$ solutions to our PDE over the spatial mesh and at some times. In particular, at each point $(x_i, t_j)$ on our grid we have $B$ solutions $u^{(b)}(x_i, t_j)$ for $b = 1, \ldots, B$, which can then be used to obtain the 2.5% and 97.5% quantiles at each point. We can therefore present confidence intervals for our solution curves, thus enabling us to quantify the uncertainty not only in the functional forms, but also in the solutions themselves.

## 3.7 Model selection

Similarly to Lagergren et al. [12], we base our model selection on the Akaike information criterion (AIC) developed by Banks and Joyner [69]. Additionally, since we have many models fit to our data in the form of bootstrap iterates, we need to consider the uncertainty in these AIC values by comparing all possible combinations. Similar methods have been used for example by Lubke et al. [70] who bootstrap AIC values to perform model selection with uncertainty.

To be particular, the formula for the AIC that we use is given by Eq 6 of Banks and Joyner [69]. If we have a model with $a$ delay parameters, $d$ diffusion parameters, and $r$ reaction parameters, then our AIC is given by

$$\text{AIC} = NM \log\left(\frac{\sum_{i=1}^{N}\sum_{j=1}^{M}(\hat{u}_{ij} - u_{ij})^2}{NM}\right) + 2(a + d + r + 1), \tag{46}$$

where these residuals $\hat{u}_{ij} - u_{ij}$ are the same as those used in computing (34). Additionally, $\hat{u}_{ij}$ refers to the PDE solution which uses a sampled initial condition rather than the initial condition which uses a spline. In this work we use this AIC corrected for a small sample size, given by Eq 17 of Banks and Joyner [69]:

$$\text{AICc} = \text{AIC} + \frac{2(a + d + r + 1)(a + d + r + 2)}{NM - a - d - r}. \tag{47}$$

We also note that if we were to extend this work to heteroscedastic Gaussian processes, the AIC formula to be used would be the weighted least squares AIC given in Eq 10 of Banks and Joyner [69].

To compare models, let us first consider the case where we have no uncertainty and are only comparing single AICs across different models. Let us have $|\mathcal{M}|$ models $\mathcal{M}_1, \ldots, \mathcal{M}_{|\mathcal{M}|}$, and denote by $\text{AICc}_i$ the AICc of $\mathcal{M}_i$, $i = 1, \ldots, |\mathcal{M}|$. We then compute the minimum AIC,

$\mathrm{AICc}_{\min} = \min_{i=1}^{|\mathcal{M}|} \mathrm{AICc}_i$, and we define

$$\Delta_i = \mathrm{AICc}_i - \mathrm{AICc}_{\min}, \quad i = 1, \dots, |\mathcal{M}|, \tag{48}$$

to be the difference between the AICc of model $\mathcal{M}_i$ and the smallest AICc. Note that a smaller AICc implies a better fitting model, thus $\Delta_i$ measures how close $\mathcal{M}_i$ is to the best fitting model according to its AICc. Using some basic modifications for the criteria suggested by Burnham and Anderson [71] for interpreting $\Delta_i$, we use the following interpretations for (48):

1. If $\Delta_i \leq 3$: There is substantial evidence that $\mathcal{M}_i$ is amongst the class of best fitting models. We denote the event $\{\Delta_i \leq 3\}$ by $E_1$.

2. If $3 < \Delta_i \leq 8$: There is considerably less evidence in favour of $\mathcal{M}_i$ being amongst the best fitting models. We denote the event $\{3 < \Delta_i \leq 8\}$ by $E_2$.

3. If $\Delta_i > 8$: There is essentially no evidence that $\mathcal{M}_i$ is amongst the best fitting models. We denote the event $\{\Delta_i > 8\}$ by $E_3$.

Now suppose we have considered $|\mathcal{M}|$ models $\mathcal{M}_1, \dots, \mathcal{M}_{|\mathcal{M}|}$, and that for model $\mathcal{M}_i$ we have used $B$ bootstrap iterations. Note that we are assuming the same number of bootstrap iterations are used for each model. This allows the same samples $\mathbf{z}^{(b)}$ to be used for each model at the $b$th bootstrap iteration, enabling the comparison across models for the same sample from the Gaussian process. These bootstrap iterations correspond to $B$ samples from $\mathcal{M}_i$, $i = 1, \dots, |\mathcal{M}|$, and we denote by $\mathcal{M}_i^{(b)}$ the $i$th sample from the $b$th model, so that $i = 1, \dots, |\mathcal{M}|$ and $b = 1, \dots, B$. We denote the *AICc* of $\mathcal{M}_i^{(b)}$ by $\mathrm{AICc}_i^{(b)}$. We then compare all models together at the same bootstrap iterations, computing

$$\Delta_i^{(b)} = \mathrm{AICc}_i^{(b)} - \mathrm{AICc}_{\min}^{(b)}, \quad i = 1, \dots, |\mathcal{M}|, \ b = 1, \dots, B, \tag{49}$$

where $\mathrm{AICc}_{\min}^{(b)} = \min_{i=1}^{|\mathcal{M}|} \mathrm{AICc}_i^{(b)}$. Finally, let us define the matrix $\mathbf{P} \in \mathbb{R}^{3 \times |\mathcal{M}|}$ such that the $(i, j)$ entry gives the proportion of times that model $\mathcal{M}_i$ has been given the $j$th interpretation of $\Delta_i$. For example, if $[\mathbf{P}]_{i2} = 0.2$, then for all the comparisons $(\Delta_i^{(1)}, \dots, \Delta_i^{(B)})$, 20% of these were such that $3 < \Delta_i^{(b)} \leq 8$, meaning 20% of the time we interpreted this model $\mathcal{M}_i$ as having considerably less evidence in favour of it being in the class of optimal models. This matrix $\mathbf{P}$ gives us a way to perform model selection in terms of probabilities that each model is amongst the class of best fitting models, close to being in this class, or not close to being in this class. The columns of $\mathbf{P}$ sum to 1, but the rows need not sum to 1 since we allow for multiple models to be in the class of best fitting models. In terms of the events $E_1$, $E_2$, and $E_3$, the $i$th row gives an estimate for the probability that $E_i$ occurs, $\mathbb{P}(E_i)$, for each model for $i = 1, 2, 3$.

We note one important feature of using the AIC as we have presented in this section. In particular, interpretation of the AIC requires a clear statistical model associated with the loss function used [71]. The loss functions (33) and (34) combined as $\mathcal{L}(\boldsymbol{\theta}) = \log[\mathcal{L}_{\mathrm{PDE}}(\boldsymbol{\theta})] + \log[\mathcal{L}_{\mathrm{GLS}}(\boldsymbol{\theta})]$, as we have done in defining (35), do not correspond to a clear statistical model. Despite this, the loss function (34) does correspond to a clear statistical model, namely the same model $u_{ij} = u(x_i, t_j) + \sigma_n z_{ij}, z_{ij} \sim \mathcal{N}(0, 1)$, used for our Gaussian process (15) [12, 58]. Since our AIC computed in (46) uses exactly the same data as is used in computing $\mathcal{L}_{\mathrm{GLS}}(\boldsymbol{\theta})$, in particular using the experimental data and the numerical solution of the PDE with the model parameters, the associated statistical model is the one used in defining $\mathcal{L}_{\mathrm{GLS}}(\boldsymbol{\theta})$, namely $u_{ij} = u(x_i, t_j) + \sigma_n z_{ij}$, rather than $\mathcal{L}(\boldsymbol{\theta})$. Therefore, our interpretation of the AIC is valid.

## Supporting information

**S1 Document. Supporting information, containing additional simulation studies, figures, and discussion.**

**S1 Text. Simulation studies**. Synthetic data explorations to demonstrate the veracity of our methods.

**S1 Fig A. Simulated data and fitted Gaussian process**. Simulated data according to the model in S1 Eq B with $D(u; \boldsymbol{\beta}) = \beta_1$ and $R(u; \boldsymbol{\gamma}) = \gamma_1 u(1-u/K)$, with $\beta_1 = 301\ \mu m^2$ and $\gamma_1 = 0.044\ h^{-1}$. The lines and shaded regions represent a Gaussian process fit to the spatiotemporal data.

**S1 Fig B. Initial simulation results**. Initial simulation results for the data from model in S1 Eq B where $R(u; \boldsymbol{\gamma}) = \gamma_1 u(1-u/K)$ and we use the misspecified nonlinear diffusivity model $D(u; \boldsymbol{\gamma}) = \beta_1 + \beta_2(u/K)$. In the density plots, the dashed red vertical line is at the position of the true parameter value, and in the curve plots the dashed red curve shows the true mechanisms.

**S1 Fig C. Final simulation results**. Final simulation results after re-scaling for the data from the model in S1 Eq 5 where $R(u; \boldsymbol{\gamma}) = \gamma_1 u(1-u/K)$ and we use the misspecified nonlinear diffusivity model $D(u; \boldsymbol{\gamma}) = \beta_1 + \beta_2(u/K)$. The re-scaling used is given in S1 Eq C. In the density plots, the dashed red vertical line is at the position of the true parameter value, and in the curve plots the dashed red curve shows the true mechanisms.

**S1 Fig D. Initial simulation results**. Initial simulation results for the data from model in S1 Eq B where $R(u; \boldsymbol{\gamma}) = \gamma_1 u(1-u/K)$ and we use the correctly specified model $D(u; \boldsymbol{\gamma}) = \beta_1$. In the density plots, the dashed red vertical line is at the position of the true parameter value, and in the curve plots the dashed red curve shows the true mechanisms.

**S1 Fig E. Final simulation results**. Final simulation results after re-scaling for the data from the model in S1 Eq B where $R(u; \boldsymbol{\gamma}) = \gamma_1 u(1-u/K)$ and we use the correctly specified nonlinear diffusivity model $D(u; \boldsymbol{\gamma}) = \beta_1$. The rescaling used is $\beta_1 = 300\hat{\beta}_1$ and $\gamma_1 = 0.044\hat{\gamma}_1$. In the density plots, the dashed red vertical line is at the position of the true parameter value, and in the curve plots the dashed red curve shows the true mechanism.

**S1 Table A. Results for the two models considered**. Interval estimates, PDE errors, and model selection results for the two nonlinear diffusivity functions considered in this study. The PDE errors are computed using sampled initial conditions. Significance levels for all uncertainty intervals are 95%. For both models we use $R(u; \boldsymbol{\gamma}) = \gamma_1 u(1-u/K)$ and $T(t; \boldsymbol{\alpha}) = 1$.

**S1 Fig F. Simulated data and fitted Gaussian process**. Simulated data according to the model in S1 Eq E with $\alpha_1 = -1.5$, $\alpha_2 = 0.31\ h^{-1}$, $\beta_1 = 571\ \mu m^2\ h^{-1}$, and $\gamma_1 = 0.081\ h^{-1}$. The lines and shaded regions represent a Gaussian process fit to the spatiotemporal data.

**S1 Table B. Results for the five models considered**. Interval estimates, PDE errors, and model selection results for the five models in S1 Eq F–J considered for the data generated by the model in S1 Eq E. 100 bootstrap iterations are used for each model. The PDE errors are computed using sampled initial conditions. Significance levels for all uncertainty intervals are 95%.

**S1 Fig G. Final model results for the optimal model**. Results for the fitted model in S1 Eq G for the data generated from S1 Eq E. In the density plots, the dashed red vertical line is at the position of the true parameter value, in the curve plots the dashed red curve shows the true mechanism, and in the curve plots the different shaded regions show various values of the products of the respective mechanisms at the shown times.

**S1 Fig H. All model results for the tested models**. All PDE solutions for the fitted models in S1 Table B. The initial conditions are sampled using the Gaussian process. The correctly specified model's results are those in (b).

**S2 Text: Basis function approach**. Alternative approach to nonlinear optimisation.

**S2 Fig I. Results for the Fisher-Kolmogorov model without delay using the basic function**

**approach**. Results for parameters, functional forms, and PDE solutions for the model in S1 Eq B using the basis function approach. In the density plots, the dashed red line is at the true parameter value, and in the curve plots the dashed red curve shows the true curve.

**S3 Text: Results for scratch assay data**. Additional results treating each experimental data set separately.

**S3 Fig J. Final model results for the optimal model with 10,000 cells per well**. Results for the Fisher-Kolmogorov model with delay for the 10,000 cells per well data set from Table 1. In the curve plots, the different shaded regions show various values of the products of the respective mechanisms at the shown times. The shaded regions in the PDE plots represent the uncertainty at each point around the mean curve shown, and the points shown are the data from Jin et al. [13].

**S3 Fig K. Final model results for the optimal model with 12,000 cells per well**. Results for the Fisher-Kolmogorov model with delay for the 12,000 cells per well data set from Table 1. In the curve plots, the different shaded regions show various values of the products of the respective mechanisms at the shown times. The shaded regions in the PDE plots represent the uncertainty at each point around the mean curve shown, and the points shown are the data from Jin et al. [13].

**S3 Fig L. Final model results for the optimal model with 14,000 cells per well**. Results for the Fisher-Kolmogorov model with delay for the 14,000 cells per well data set from Table 1. In the curve plots, the different shaded regions show various values of the products of the respective mechanisms at the shown times. The shaded regions in the PDE plots represent the uncertainty at each point around the mean curve shown, and the points shown are the data from Jin et al. [13].

**S3 Fig M. Final model results for the optimal model with 16,000 cells per well**. Results for the Fisher-Kolmogorov model with delay for the 16,000 cells per well data set from Table 1. In the curve plots, the different shaded regions show various values of the products of the respective mechanisms at the shown times. The shaded regions in the PDE plots represent the uncertainty at each point around the mean curve shown, and the points shown are the data from Jin et al. [13].

**S3 Fig N. Final model results for the optimal model with 18,000 cells per well**. Results for the Fisher-Kolmogorov model with delay for the 18,000 cells per well data set from Table 1. In the curve plots, the different shaded regions show various values of the products of the respective mechanisms at the shown times. The shaded regions in the PDE plots represent the uncertainty at each point around the mean curve shown, and the points shown are the data from Jin et al. [13].

**S3 Fig O. Final model results for the optimal model with 20,000 cells per well**. Results for the Fisher-Kolmogorov model with delay for the 20,000 cells per well data set from Table 1. In the curve plots, the different shaded regions show various values of the products of the respective mechanisms at the shown times. The shaded regions in the PDE plots represent the uncertainty at each point around the mean curve shown, and the points shown are the data from Jin et al. [13].

**S3 Fig P. Comparisons to Lagergren et al.'s results with 10,000 cells per well**. Results for the generalised Porous-FKPP model with delay for the 10,000 cells per well data set from Table 1. In the curve plots, the different shaded regions show various values of the products of the respective mechanisms at the shown times. The shaded regions in the PDE plots represent the uncertainty at each point around the mean curve shown, and the points shown are the data from Jin et al. [13].

**S3 Fig Q. Comparisons to Lagergren et al.'s results with 12,000 cells per well**. Results for the generalised Porous-FKPP model with delay for the 12,000 cells per well data set from Table 1. In

the curve plots, the different shaded regions show various values of the products of the respective mechanisms at the shown times. The shaded regions in the PDE plots represent the uncertainty at each point around the mean curve shown, and the points shown are the data from Jin et al. [13].

**S3 Fig R. Comparisons to Lagergren et al.'s results with 14,000 cells per well**. Results for the generalised Porous-FKPP model with delay for the 14,000 cells per well data set from Table 1. In the curve plots, the different shaded regions show various values of the products of the respective mechanisms at the shown times. The shaded regions in the PDE plots represent the uncertainty at each point around the mean curve shown, and the points shown are the data from Jin et al. [13].

**S3 Fig S. Comparisons to Lagergren et al.'s results with 16,000 cells per well**. Results for the generalised Porous-FKPP model with delay for the 16,000 cells per well data set from Table 1. In the curve plots, the different shaded regions show various values of the products of the respective mechanisms at the shown times. The shaded regions in the PDE plots represent the uncertainty at each point around the mean curve shown, and the points shown are the data from Jin et al. [13].

**S3 Fig T. Comparisons to Lagergren et al.'s results with 18,000 cells per well**. Results for the generalised Porous-FKPP model with delay for the 18,000 cells per well data set from Table 1. In the curve plots, the different shaded regions show various values of the products of the respective mechanisms at the shown times. The shaded regions in the PDE plots represent the uncertainty at each point around the mean curve shown, and the points shown are the data from Jin et al. [13].

**S3 Fig U. Comparisons to Lagergren et al.'s results with 20,000 cells per well**. Results for the generalised Porous-FKPP model with delay for the 20,000 cells per well data set from Table 1. In the curve plots, the different shaded regions show various values of the products of the respective mechanisms at the shown times. The shaded regions in the PDE plots represent the uncertainty at each point around the mean curve shown, and the points shown are the data from Jin et al. [13].

**S4 Text: Data thresholding**. Additional improved results by thresholding data beyond certain limits.

**S4 Table C. Results for data thresholding**. Results for pairs of ($\delta u$, $\delta t$). The errors reported are mean percentage errors, rounded to three decimal places, when simulating the Fisher-Kolmogorov model in S1 Eq B as described in S1 Text. The bold values show the minimum error in the column, and the underlined values show the minimum error in the row. The blue cell shows the minimum error in the whole table.

**S4 Table D. Results for data thresholding**. Results for pairs of ($\delta u$, $\delta t$). The errors reported are mean percentage errors, rounded to three decimal places, when simulating the Fisher-Kolmogorov model with delay as described in S1 Text. The bold values show the minimum error in the column, and the underlined values show the minimum error in the row. The blue cell shows the minimum error in the whole table.

**S5 Text: PDE truncation error**. Additional results exploring the impact of using different spatial discretisations for the numerical PDE solution.

**S5 Table E. Model selection results for the five models considered with $N_P$ = 500**. Interval estimates, PDE errors, and model selection results for the models in Eq 2–6 when applied to each data set from Jin et al. [13]. The time taken to produce these results is 31 hours and 40 minutes.

**S5 Table F. Model selection results for the five models considered with $N_P$ = 250**. Interval estimates, PDE errors, and model selection results for the models in Eq 2–6 when applied to each data set from Jin et al. [13]. The time taken to produce these results is 4 hours and 37 minutes.

**S5 Table G. Model selection results for the five models considered with $N_P$ = 125**. Interval

estimates, PDE errors, and model selection results for the models in Eq 2–6 when applied to each data set from Jin et al. [13]. The time taken to produce these results is 1 hour and 33 minutes.

**S5 Table H. Model selection results for the five models considered with** $N_p$ = 50. Interval estimates, PDE errors, and model selection results for the models in Eq 2–6 when applied to each data set from Jin et al. [13]. The time taken to produce these results is 56 minutes and 7 seconds.

**S5 Fig V. Coefficient comparisons for different grid sizes**. Comparing coefficients from model 2 for each data set, using different values for $N_p$. The lines shown are the 95% confidence intervals for the corresponding coefficient. The coefficients are all scaled by a value $\hat{\theta}$, e.g. $\alpha_1$ is scaled by $\hat{\alpha}_1$, where $\hat{\theta}$ is the mean value of the coefficient from the results using $N_p$ = 500 for that data set, and similarly for the PDE errors.

**S6 Text. Notation**. Summary of mathematical notation.

**S6 Table I. Grid and density notation**. Notation used for several grids and for labelling the density values.

**S6 Table J. Gaussian process notation**. Notation used for the Gaussian processes.

**S6 Table K. Equation learning, bootstrapping, and model selection notation**. Notation used for equation learning, bootstrapping, and model selection.

(PDF)

## Author Contributions

**Conceptualization:** Daniel J. VandenHeuvel, Christopher Drovandi, Matthew J. Simpson.

**Data curation:** Daniel J. VandenHeuvel.

**Formal analysis:** Daniel J. VandenHeuvel, Christopher Drovandi, Matthew J. Simpson.

**Funding acquisition:** Matthew J. Simpson.

**Investigation:** Daniel J. VandenHeuvel, Matthew J. Simpson.

**Methodology:** Daniel J. VandenHeuvel, Christopher Drovandi.

**Project administration:** Matthew J. Simpson.

**Resources:** Daniel J. VandenHeuvel.

**Software:** Daniel J. VandenHeuvel.

**Supervision:** Christopher Drovandi, Matthew J. Simpson.

**Visualization:** Daniel J. VandenHeuvel.

**Writing – original draft:** Daniel J. VandenHeuvel.

**Writing – review & editing:** Daniel J. VandenHeuvel, Christopher Drovandi, Matthew J. Simpson.

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
