## [Decision Letter · Decision Letter 0]

6 Jul 2022

Dear Professor Simpson,

Thank you very much for submitting your manuscript "Computationally efficient mechanism discovery for cell invasion with uncertainty quantification." for consideration at PLOS Computational Biology.

As with all papers reviewed by the journal, your manuscript was reviewed by members of the editorial board and by several independent reviewers. In light of the reviews (below this email), we would like to invite the resubmission of a significantly-revised version that takes into account the reviewers' comments.

We cannot make any decision about publication until we have seen the revised manuscript and your response to the reviewers' comments. Your revised manuscript is also likely to be sent to reviewers for further evaluation.

Sincerely,

Philip K Maini

Associate Editor

PLOS Computational Biology

Mark Alber

Deputy Editor

PLOS Computational Biology

Reviewer's Responses to Questions

**Comments to the Authors:**

Reviewer #1: Uploaded as an attachment

Reviewer #2: This study proposes a framework for discovering biological mechanisms using equation learning, with the additional benefit of providing uncertainty quantification estimates. It specifically applies this framework to PDE models of cell invasion scratch assay experiments, where they perform model selection to study mechanisms such as delay, migration, and proliferation. The framework is based on using Gaussian processes combined with bootstrapping, as opposed to neural networks or Bayesian methods previously employed for this type of learning. The method seems to have some advantages in terms of dealing with noisy and sparse data, being parallelizable, and enabling uncertainty quantification. The manuscript is largely well-written, and I recommend the following revisions:

1. In section 3.1, the authors discuss their procedure for choosing a carrying capacity K for their study. In the Discussion, they briefly mention adding K to the list of parameters to be estimated. Do you have any intuition on how adding such a parameter may make the framework more computationally extensive, or less robust?

2. This may be discussed in the cited publications, but what is the intuition behind the delay term for these experiments?

3. Section 3.3.1 could provide a bit more background on the synthetic data studies in the supplement, especially since they find that an iterative approach to parameter re-scaling may be necessary.

4. In section 3.3.2, the authors start describing some ways to quantify error based on bootstraps sampled from the Gaussian process. This becomes clearer once one reads the Methods, but I wonder if moving up some of the Methods or providing an overview of the framework before these sections would make these Results sections more accessible.

5. In the long paragraph on page 13, it would be helpful to break down several of the sentences into shorter ones to make the discussion of the hypothesis testing and differences between datasets more readable.

6. In the second paragraph on page 13, it was unclear why the previous study (Lagergren et al.) constrained the diffusivity function to be increasing. If there is a biological motivation for this, wouldn’t we expect to see the same trend with this new framework?

7. Are there parameter identifiability issues with any of the models considered here? How does this framework connect to results discussed in the authors’ 2020 study on “Practical parameter identifiability for spatio-temporal models of cell invasion” for the same datasets?

8. Does the framework extend to learning biological mechanisms in systems of partial differential equations models?

9. In eq. (14), the authors could extend the second row of matrix X to make it clearer that t_1 is repeated N times, then t_2, etc.

10. On page 20 before eq. (17), subscript notation for u_* is used. Is this intended to be the same notation as in eq. (15)? This could be made clearer.

11. In section 4.3, are there numerical concerns that matrix sigma is not symmetric positive definite?

12. The authors could provide more intuition on the two loss functions introduced on page 23. For instance, how does the first loss function, which is based on the methodology proposed here (draws from the fitted Gaussian process), improve the optimization?

13. On top of page 29, “jth sample” should probably be “ith” sample.

14. In the example on page 29, should j=2 to correspond to the second interpretation for the AIC score differences?

Minor edits:

- Page 4, sentence starting with “Images of the experiment” needs a verb.

- Page 5, replace “are showing” with “are shown”.

- Page 22, typo “Cholesky factory”.

- The use of “say” on page 24 seems a bit informal and vague.

- Page 28, typo “the its AICc”.

Reviewer #3: The manuscript considers a specific application of quite a generic task: how to select, amongst a set of candidate models, which one is best supported by the data. It adopts a bootstrap (simulation-based) approach, aimed at attributing a notion of uncertainty to the inferred model parameters and selected models. Much is nicely written, though I frequently found myself struggling to follow the narrative and the big picture. There is extensive details about *what* was done in terms of methodological choices, but little explanation/justification about *why* these choices were made and how they impact the statistical properties of the inference. Consequently, it is tricky to follow how the contribution relates to the literature and whether the statistical properties of the approach make it attractive.

Major points

- The paper omits to consider classical approaches to model selection. S2 cites recent "Mechanism discovery"/ "equation learning" papers and mentions these, and more classical Bayesian approaches, as alternative methodological approaches to the ones being presented. I do not understand, though, why classical approaches to model selection, such as the likelihood-ratio test (LRT, for nested models) and information-criteria based methods (for not necessarily nested models) [ref 62] are not mentioned here and used for comparison throughout the manuscript. These methods date back decades and seem closest in spirit to the approach being taken here. Note that although the LRT is restricted to comparisons of nested models, it is a method of model selection that incorporates uncertainty quantification, and it is very cheap (requiring only two evaluations of the likelihood).

- "Uncertainty quantification" is emphasised throughout as a key contribution of the approach, without explanation of what this means. "Uncertainty" is an overloaded phrase in statistics, and the notion of uncertainty (and the underlying probability, whether frequentist or subjective) is only useful if it is precisely prescribed what is meant by it. This is not mere statistical fussiness: it is central to understanding the role of GPs in the method in this manuscript, and what the resulting, and currently ambiguous, "uncertainty intervals" and "goodness of fit probabilities" actually mean. I *think* (though I'm not sure) that "uncertainty intervals" for fitted parameters are just classical "confidence intervals" interpreted in the frequentist sense. The "goodness of fit" probabilities are more non-standard (something I'll come back to below)

- GPs are central to the approach, but the role they play isn't fully explained. My understanding (with apologies if I have misunderstood) is that the approach and rationale is as follows:

1. The GP is fitted to the experimental data and then simulated from.

2. The GP (sampled at discrete intervals mimicking experimental resolution, possibly plus additive noise) is being considered as a surrogate for the data-generating process.

3. Therefore, bootstrapping (repeatedly simulating a data set from the GP model) mimics repeated collection of novel data sets.

4. For each bootstrap realisation of the data set, multiple candidate models are fitted. This provides point estimates of model parameters, and rankings of the suitability of candidate models (according to some chosen criterion).

5. Repeating over multiple bootstrap realisations provides a sampling distribution for the parameter estimates and model rankings. These are interpreted in the frequentist sense, and hence can be used to compute confidence intervals.

If this interpretation is correct then (i) I think the manuscript would benefit from spelling out such a rationale clearly, (ii) this makes precise that the "uncertainty intervals" in the results really are just frequentist confidence intervals, and should be called that, (iii) it makes clear that point 2 is absolutely crucial: the entire notion of uncertainty, and reliability of inference, hinges on the GP being a reasonable surrogate for the data-generating process. Too much (resp. too little) uncertainty on the simulated data means too much (resp. too little) uncertainty in the inferred results. If the GP over- or under-smooths then there could be all kinds of undesirable consequences for the model inference (which after all is geared here towards selecting different physical models for "smoothing" the observable variable).

Is the data-generating process really being carefully mimicked by the GP? On p6, "the uncertainty intervals do not precisely capture the data points...If we were to capture the data itself, an extra term could be added to the variance estimates" makes it sound as though the authors have in mind a model such as

y_i = f(x_i) + \\epsilon_i (1)

and they are simulating E(y_i) (with some variability between GP realisation, though it is unclear to me how this variability is interpreted) rather than y_i, which would be the natural choice if my interpretation 1-5 above were accurate. Is it even necessary to use GP at all? A common bootstrap approach in a setting like this is to "bootstrap the residuals", i.e., compute a non-parametric estimate \\hat{f} of f, compute the residuals r_i = \\hat{f}(x_i) - y_i, then simulate bootstrapped data as y^*_i = \\hat{f}(x_i) + r^*_i where each r^*_i is sampled uniformly at random from the set {r_i}. This feels a bit more direct, and helps to make sure the variability in the simulated/bootstrap data is comparable to the variability in the experiment data - but are there reasons why a GP is preferable to this?

- "Computational efficiency" is another property of the approach that is emphasised, e.g. in the title but not properly explained. This needs to be carefully spelled out: in various methods including the one you propose, which steps are computationally costly and how many times must they be performed? How does the proposed method sidestep, or require fewer of, the costly steps?

Other points

- p1 What does "efficient" mean?

- On p2 it is unclear to me what the authors mean by "Identifying appropriate terms in a mathematical model is traditionally accomplished through empirical reasoning. Model discovery methods instead ... ". It feels as though "empirical reasoning" encompasses pretty much everything. What are the traditional approaches? Aren't the cited "model discovery methods" just a type of empirical reasoning? Mention here the likelihood-ratio test and IC-based model selection approaches, and how the work in the manuscript relates to them.

- p2 "realistic models are obtained" needs clarification - in what sense "realistic"? (Aren't all models wrong?)

- p3 top: the role of the GP needs explanation here. Why choose a GP, and *how* does it enable UQ?

- p3 "sparse data" in what sense? (Compared to many biological data sets, the spatial and temporal resolution seem to be very high here)

- p3 "Bayesian methods" - needs more explanation. (I'm guessing this refers to methods involving statistical model parameterised in terms of the solution of a DE involving parameters, in which inference is based on the likelihood function of this statistical model - and even if so, introducing the idea here would be helpful for later)

- p3 "[compared with the Bayesian approach] our approach ... allows us to enable a candidate library of terms in the PDE model similar to [equation learning]". But the number of models considered (p9) is only very small. Why couldn't a similar number be compared in the Bayesian approach? Isn't the compute time per evaluation of the likelihood essentially the same, because both entail solving the PDE (presumably the slow bit)? Or is the idea here (and of "computational efficiency" throughout) just that the likelihood needs to be evaluated fewer times, i.e. the # of bootstrap samples is fewer than the number of MCMC iterations?

- p4 What did Jin et al do to analyse these data? What was their approach and their conclusions? How do these relate to the approach and conclusion of the current manuscript?

- p6 Need explanation of *why* the GP is being introduced. What role does it play, and why that choice? What precisely does the "uncertainty interval" mean?

- p8 at this point, it is unclear how the GP facilitates "learning the mechanisms", what you will do with the repeatedly sampled values and the GP derivatives, and what is the optimisation problem that's being referring to.

- p9, eqn (2) - (6): no need to repeat the eqn for R if it is identical in each case.

- p9 eqn (4), (5), i.e. models 3 and 4: it would seem more natural to me to relabel that beta_1 as beta_2. Then it is clear that the "beta" in models 3 and 4 plays a different role to the beta_1 in models 1 and 2, and it makes the model nesting clearer - for example, that model 4 is a special case of model 5.

- p9 "We will fit these models to each dataset" needs at least brief explanation here (not just a reference to the methods) especially as the fitting criterion is quite non-standard. Why choose such an unusual loss function (defined later on p23), in terms of the derivative of the state variable, rather than in terms of the state variable itself?

- p9 "... then compare the using Akaike IC." What is the implied statistical model associated with the unusual loss function? Doesn't the interpretation of the IC depend on the "likelihood" component of the IC relating to an assumed statistical model, rather than an arbitrarily chosen loss?

- p11 Probabilities P(E1), P(E2), P(E3). "Prob that the model is in the class of best-fitting models", "that there is only some evidence", "that there is no evidence" - again can't be understood unless the reader jumps forwards to the methods. In what statistical sense are these probabilities to be interpreted?

- p12 Table 1 - it's a bit confusing to have the label "Models" then column labels (2)-(6) which refers to the equation numbers not the model numbers (which differ by 1).

-p12 Table 1 - It would be helpful to have some kind of point estimate for the parameter values, e.g. a modal value from the sampling distributions. An important comparator to include parameter estimates computed as maximum likelihood estimates (MLEs) from the data themselves, rather than the GP surrogate (since this would be a conventional starting point most would take when faced with a data such as this, and a model to fit to the data). Would the point estimates using bootstrap sampling be close to the MLE? One would hope that the MLEs would lie well inside the "uncertainty intervals" presented in this table, but there is no reason to suppose they would in general, especially if the GP isn't working so well as a surrogate for the data-generating process (e.g. if it oversmooths and/or involves too little variability). Once each model is fitted by MLE then the models can be compared by LRT or AIC too - would Model 2 still be the favoured one?

**Have the authors made all data and (if applicable) computational code underlying the findings in their manuscript fully available?**

Reviewer #1: Yes

Reviewer #2: Yes

Reviewer #3: Yes

PLOS authors have the option to publish the peer review history of their article (what does this mean?). If published, this will include your full peer review and any attached files.

Reviewer #1: **Yes: **Ben Lambert

Reviewer #2: No

Reviewer #3: No
---

## [Decision Letter · Decision Letter 1]

13 Sep 2022

Dear Professor Simpson,

Thank you very much for submitting your manuscript "Computationally efficient mechanism discovery for cell invasion with uncertainty quantification." for consideration at PLOS Computational Biology. As with all papers reviewed by the journal, your manuscript was reviewed by members of the editorial board and by several independent reviewers. The reviewers appreciated the attention to an important topic. Based on the reviews, we are likely to accept this manuscript for publication, providing that you modify the manuscript according to the review recommendations.

Sincerely,

Philip K Maini

Academic Editor

PLOS Computational Biology

Mark Alber

Section Editor

PLOS Computational Biology

[LINK]

Reviewer's Responses to Questions

**Comments to the Authors:**

Reviewer #1: I thank the authors for their detailed resubmission and responses, which addressed all of my concerns. As an aside, I found the work that they have subsequently done on the impact of grid sizes on estimates really quite dramatic -- that's quite a speed up -- and it's interesting how robust the estimates are to these numerics.

Reviewer #2: The authors have addressed all my questions and suggestions in the revised manuscript.

Reviewer #3: The authors have clearly gone to good lengths to address the comments from me and the other reviewers.

I do have a remaining point relating to how the uncertainty quantification should be interpreted (which it seems key to be clear on given that this is a central focus of the manuscript). p10 of the revised manuscript directly addresses my points 4 and 5 along with their response "We do not attempt to mimic the precise data-generating process with our Gaussian process. Instead, we use the Gaussian process as a surrogate for the dynamical system that best describes the de-noised data-generating process". This is clearly explained but I'm still not clear on the rationale for that choice. Usually a parametric bootstrap seeks to mimic the data-generating process as closely as possible (including the noise structure of the observed data) so that, e.g., the CIs have a precise interpretation as uncertainty in the inferred parameters owing to sampling variability, i.e. having a finite number of noisy data points. If the parametric bootstrap is mimicking a de-noised process (albeit with some variability) then can the authors clarify how are CIs, and other inference, should instead be interpreted?

**Have the authors made all data and (if applicable) computational code underlying the findings in their manuscript fully available?**

Reviewer #1: Yes

Reviewer #2: Yes

Reviewer #3: Yes

PLOS authors have the option to publish the peer review history of their article (what does this mean?). If published, this will include your full peer review and any attached files.

Reviewer #1: **Yes: **Ben Lambert

Reviewer #2: No

Reviewer #3: No

Figure Files:

Data Requirements:

Reproducibility:

References:

---

## [Editor Report · Decision Letter 2]

23 Sep 2022

Dear Professor Simpson,

We are pleased to inform you that your manuscript 'Computationally efficient mechanism discovery for cell invasion with uncertainty quantification.' has been provisionally accepted for publication in PLOS Computational Biology.

Best regards,

Philip K Maini

Academic Editor

PLOS Computational Biology

Mark Alber

Section Editor

PLOS Computational Biology

---

## [Editor Report · Acceptance letter]

9 Nov 2022

PCOMPBIOL-D-22-00752R2 

Computationally efficient mechanism discovery for cell invasion with uncertainty quantification.

Dear Dr Simpson,

I am pleased to inform you that your manuscript has been formally accepted for publication in PLOS Computational Biology. Your manuscript is now with our production department and you will be notified of the publication date in due course.

With kind regards,

Zsanett Szabo
